# ALISTA: ANALYTIC WEIGHTS ARE AS GOOD AS LEARNED WEIGHTS IN LISTA

**Jialin Liu**[*]
Department of Mathematics
University of California, Los Angeles
liujl11@math.ucla.edu

**Xiaohan Chen**[*]
Department of Computer Science and Engineering
Texas A&M University
chernxh@tamu.edu

**Zhangyang Wang**
Department of Computer Science and Engineering
Texas A&M University
atlaswang@tamu.edu

**Wotao Yin**
Department of Mathematics
University of California, Los Angeles
wotaoyin@math.ucla.edu

## ABSTRACT

Deep neural networks based on unfolding an iterative algorithm, for example, LISTA (learned iterative shrinkage thresholding algorithm), have been an empirical success for sparse signal recovery. The weights of these neural networks are currently determined by data-driven "black-box" training. In this work, we propose Analytic LISTA (ALISTA), where the weight matrix in LISTA is computed as the solution to a data-free optimization problem, leaving only the stepsize and threshold parameters to data-driven learning. This significantly simplifies the training. Specifically, the data-free optimization problem is based on coherence minimization. We show our ALISTA retains the optimal linear convergence proved in (Chen et al., 2018) and has a performance comparable to LISTA. Furthermore, we extend ALISTA to convolutional linear operators, again determined in a data-free manner. We also propose a feed-forward framework that combines the data-free optimization and ALISTA networks from end to end, one that can be jointly trained to gain robustness to small perturbations in the encoding model.

## 1 INTRODUCTION

Sparse vector recovery, or sparse coding, is a classical problem in source coding, signal reconstruction, pattern recognition and feature selection. There is an unknown sparse vector $\mathbf{x}^* = [x_1^*, \cdots, x_M^*]^T \in \mathbb{R}^M$. We observe its noisy linear measurements:

$$\mathbf{b} = \sum_{m=1}^{M} \mathbf{d}_m x_m^* + \varepsilon = \mathbf{D}\mathbf{x}^* + \varepsilon, \tag{1}$$

where $\mathbf{b} \in \mathbb{R}^N$, $\mathbf{D} = [\mathbf{d}_1, \cdots, \mathbf{d}_M] \in \mathbb{R}^{N \times M}$ is the *dictionary*, and $\varepsilon \in \mathbb{R}^N$ is additive Gaussian white noise. For simplicity, each column of $\mathbf{D}$, named as a *dictionary kernel*, is normalized, that is, $\|\mathbf{d}_m\|_2 = \|\mathbf{D}_{:,m}\|_2 = 1$, $m = 1, 2, \cdots, M$. Typically, we have $N \ll M$, so Equation (1) is an under-determined system.

However, when $\mathbf{x}^*$ is sufficiently sparse, it can be recovered faithfully. A popular approach is to solve the LASSO problem below (where $\lambda$ is a scalar):

$$\underset{\mathbf{x}}{\text{minimize}} \frac{1}{2}\|\mathbf{b} - \mathbf{D}\mathbf{x}\|_2^2 + \lambda\|\mathbf{x}\|_1 \tag{2}$$

using iterative algorithms such as the iterative shrinkage thresholding algorithm (ISTA):

$$\mathbf{x}^{(k+1)} = \eta_{\lambda/L}\Big(\mathbf{x}^{(k)} + \frac{1}{L}\mathbf{D}^T(\mathbf{b} - \mathbf{D}\mathbf{x}^{(k)})\Big), \quad k = 0, 1, 2, \dots \tag{3}$$

---

[*]These authors contributed equally and are listed alphabetically.

where $\eta_\theta$ is the soft-thresholding function[1] and $L$ is usually taken as the largest eigenvalue of $\mathbf{D}^T\mathbf{D}$.

Inspired by ISTA, the authors of (Gregor & LeCun, 2010) proposed to learn the weights in the matrices in ISTA rather than fixing them. Their methods is called Learned ISTA (LISTA) and resembles a recurrent neural network (RNN). If the iteration is truncated to $K$ iterations, LISTA becomes a $K$-layer feed-forward neural network with side connections. Specifically, LISTA is:

$$\mathbf{x}^{(k+1)} = \eta_{\theta^{(k)}}(\mathbf{W}_1^{(k)}\mathbf{b} + \mathbf{W}_2^{(k)}\mathbf{x}^{(k)}), \quad k = 0, 1, \cdots, K-1. \tag{4}$$

If we set $\mathbf{W}_1^{(k)} \equiv \frac{1}{L}\mathbf{D}^T$, $\mathbf{W}_2^{(k)} \equiv \mathbf{I} - \frac{1}{L}\mathbf{D}^T\mathbf{D}$, $\theta^{(k)} \equiv \frac{1}{L}\lambda$, then LISTA recovers ISTA. Given each pair of sparse vector and its noisy measurements $(\mathbf{x}^*, \mathbf{b})$, applying (4) from some initial point $\mathbf{x}^{(0)}$ and using $\mathbf{b}$ as the input yields $\mathbf{x}^{(k)}$. Our goal is to choose the parameters $\Theta = \{\mathbf{W}_1^{(k)}, \mathbf{W}_w^{(k)}, \theta^{(k)}\}_{k=0,1,\ldots,K-1}$ such that $\mathbf{x}^{(k)}$ is close to $\mathbf{x}^*$ for all sparse $\mathbf{x}^*$ following some distribution $\mathcal{P}$. Therefore, given the distribution $\mathcal{P}$, all parameters in $\Theta$ are subject to learning:

$$\underset{\Theta}{\text{minimize}} \, \mathbb{E}_{\mathbf{x}^*, \mathbf{b} \sim \mathcal{P}} \left\| \mathbf{x}^{(K)}\left(\Theta, \mathbf{b}, \mathbf{x}^{(0)}\right) - \mathbf{x}^* \right\|_2^2. \tag{5}$$

This problem is approximately solved over a training dataset $\{(\mathbf{x}_i^*, \mathbf{b}_i)\}_{i=1}^N$ sampled from $\mathcal{P}$.

Many empirical results, e.g., (Gregor & LeCun, 2010; Sprechmann et al., 2015; Wang et al., 2016b), show that a trained $K$-layer LISTA (with $K$ usually set to $10 \sim 20$) or its variants can generalize more than well to unseen samples $(\mathbf{x}', \mathbf{b}')$ from the same distribution and recover $\mathbf{x}'$ from $\mathbf{b}'$ to the same accuracy within one or two order-of-magnitude fewer iterations than the original ISTA. Additionally, the accuracies of the outputs $\{\mathbf{x}^{(k)}\}$ of the layers $k = 1, .., K$ gradually improve. However, such networks will generalize worse when the input deviates from the training distribution (e.g., when $\mathbf{D}$ varies), in contrast to the classical iterative algorithms such as ISTA that are training-free and thus agnostic to the input distribution. The Analysis-Synthesis model (Rubinstein & Elad, 2014; Yang et al., 2016) could also be viewed as a special LISTA model with only one layer ($K = 1$).

More recently, the *convolutional sparse coding* (CSC), an extension of the sparse coding (1), gains increasingly attention in the machine learning area. (Sreter & Giryes, 2018) showed that the CSC could be similarly approximated and accelerated by a LISTA-type feed-forward network. (Tolooshams et al., 2018) designed a structure of sparse auto-encoder inspired by multi-layer CSC. (Papyan et al., 2016; Sulam et al., 2017) also revealed CSC as a potentially useful tool for understanding general convolutional neural networks (CNNs).

## 1.1 RELATED WORK

Despite the empirical success (Sprechmann et al., 2015; Wang et al., 2016a;b;c;d; Zhang & Ghanem, 2018; Zhou et al., 2018; Ito et al., 2018) in constructing fast trainable regressors for approximating iterative sparse solvers, the theoretical understanding of such approximations remains limited.

A handful of recent works have been investigating the theory of LISTA. (Moreau & Bruna, 2017) re-factorized the Gram matrix of dictionary, by trying to nearly diagonalize the Gram matrix with a basis, subject to a small $\ell_1$ perturbation. They thus re-parameterized LISTA a new factorized architecture that achieved similar acceleration gain to LISTA, hence ending up with an "indirect" proof. They concluded that LISTA can converge faster than ISTA, but still sublinearly. (Giryes et al., 2018) interpreted LISTA as a projected gradient descent descent (PGD) where the projection step was inaccurate, which enables a trade-off between approximation error and convergence speed. The latest work (Chen et al., 2018) presented the more related results to ours: they introduced necessary conditions for the LISTA weight structure in order to achieve asymptotic **linear** convergence of LISTA, which also proved to be a theoretical convergence rate **upper bound**. They also introduced a thresholding scheme for practically improving the convergence speed. Note that, none of the above works extended their discussions to CSC and its similar LISTA-type architectures.

Several other works examined the theoretical properties of some sibling architectures to LISTA. (Xin et al., 2016) studied the model proposed by (Wang et al., 2016b), which unfolded/truncated the iterative hard thresholding (IHT) algorithm instead of ISTA, for approximating the solution to $\ell_0$-minimization. They showed that the learnable fast regressor can be obtained by using a transformed dictionary with improved restricted isometry property (RIP). However, their discussions are not

---

[1]Soft- thresholding function is defined in a component-wise way: $\eta_\theta(\mathbf{x}) = \text{sign}(\mathbf{x}) \max(0, |\mathbf{x}| - \theta)$

applicable to LISTA directly, although IHT is linearly convergent (Blumensath & Davies, 2009) under rather strong assumptions. Their discussions were also limited to linear sparse coding and resulting fully-connected networks only. (Borgerding et al., 2017; Metzler et al., 2017) studied a similar learning-based model inspired from another LASSO solver, called approximated message passing (AMP). (Borgerding et al., 2017) showed the MMSE-optimality of an AMP-inspired model, but not accompanied with any convergence rate result. Also, the popular assumption in analyzing AMP algorithms (called "state evolution") does not hold when analyzing ISTA.

### 1.2 MOTIVATION AND CONTRIBUTIONS

This paper presents multi-fold contributions in advancing the theoretical understanding of LISTA, beyond state-of-the-art results. Firstly, we show that the layer-wise weights in LISTA **need not being learned from data**. That is based on decoupling LISTA training into a *data-free* analytic optimization stage followed by a lighter-weight *data-driven* learning stage without compromising the optimal linear convergence rate proved in (Chen et al., 2018). We establish a minimum-coherence criterion between the desired LISTA weights and the dictionary $\mathbf{D}$, which leads to an efficient algorithm that can analytically solve the former from the latter, independent of the distribution of $\mathbf{x}$. The data-driven training is then reduced to learning layer-wise step sizes and thresholds only, which will fit the distribution of $\mathbf{x}$. The new scheme, called Analytic LISTA (**ALISTA**), provides important insights into the working mechanism of LISTA. Experiments shows ALISTA to perform comparably with previous LISTA models (Gregor & LeCun, 2010; Chen et al., 2018) with much lighter-weight training. Then, we extend the above discussions and conclusions to CSC, and introduce an efficient algorithm to solve the convolutional version of coherence minimization. Further, we introduce a new robust LISTA learning scheme benefiting from the decoupled structure, by adding perturbations to $\mathbf{D}$ during training. The resulting model is shown to possess much stronger robustness when the input distribution varies, even when $\mathbf{D}$ changes to some extent, compared to classical LISTA models that learn to (over-)fit one specific $\mathbf{D}$.

## 2 ANALYTIC LISTA: CALCULATING WEIGHTS WITHOUT TRAINING

We theoretically analyze the LISTA-CPSS model defined in (Chen et al., 2018):

$$\mathbf{x}^{(k+1)} = \eta_{\theta^{(k)}}\Big(\mathbf{x}^{(k)} - (\mathbf{W}^{(k)})^T(\mathbf{D}\mathbf{x}^{(k)} - \mathbf{b})\Big), \tag{6}$$

where $\mathbf{W}^{(k)} = [\mathbf{w}_1^{(k)}, \cdots, \mathbf{w}_M^{(k)}] \in \mathbb{R}^{N \times M}$ is a linear operator with the same dimensionality with $\mathbf{D}$, $\mathbf{x}^{(k)} = [x_1^{(k)}, \cdots, x_M^{(k)}]$ is the $k^{\text{th}}$ layer node. In (6), $\Theta = \{\mathbf{W}^{(k)}, \theta^{(k)}\}_k$ are parameters to train. Model (6) can be derived from (4) with $\mathbf{W}_1^{(k)} = (\mathbf{W}^{(k)})^T$, $\mathbf{W}_2^{(k)} = \mathbf{I} - \mathbf{W}_1^{(k)}\mathbf{D}$. (Chen et al., 2018) showed that (6) has the same representation capability with (4) on the sparse recovery problem, with a specifically light weight structure.

Our theoretical analysis will further define and establish properties of "good" parameters $\Theta$ in (6), and then discuss how to analytically compute those good parameters rather than relying solely on black-box training. In this way, the LISTA model could be further significantly simplified, with little performance loss. The proofs of all the theorems in this paper are provided in the appendix.

### 2.1 RECOVERY ERROR UPPER BOUND

We start with an assumption on the "ground truth" signal $\mathbf{x}^*$ and the noise $\varepsilon$.

**Assumption 1** (Basic assumptions). *Signal $\mathbf{x}^*$ is sampled from the following set:*

$$\mathbf{x}^* \in \mathcal{X}(B, s) \triangleq \Big\{\mathbf{x}^* \Big| |x_i^*| \leq B, \forall i, \ \|\mathbf{x}^*\|_0 \leq s\Big\}. \tag{7}$$

*In other words, $\mathbf{x}^*$ is bounded and $s$-sparse[2] ($s \geq 2$). Furthermore, we assume $\varepsilon = \mathbf{0}$.*

The zero-noise assumption is for simplicity of the proofs. Our experiments will show that our models are robust to noisy cases.

The *mutual coherence* of the dictionary $\mathbf{D}$ is a significant concept in compressive sensing (Donoho & Elad, 2003; Elad, 2007; Lu et al., 2018). A dictionary with small coherence possesses better sparse recovery performance. Motivated by this point, we introduce the following definition.

---

[2] A signal is $s$-sparse if it has no more than $s$ non-zero entries.

**Definition 1.** *Given* $\mathbf{D} \in \mathbb{R}^{N \times M}$ *with each of its column normalized, we define the generalized mutual coherence:*

$$\tilde{\mu}(\mathbf{D}) = \inf_{\substack{\mathbf{W} \in \mathbb{R}^{N \times M} \\ (\mathbf{W}_{:,i})^T \mathbf{D}_{:,i}=1, 1 \leq i \leq M}} \left\{ \max_{\substack{i \neq j \\ 1 \leq i,j \leq M}} (\mathbf{W}_{:,i})^T \mathbf{D}_{:,j} \right\}. \tag{8}$$

*Additionally, We define* $\mathcal{W}(\mathbf{D}) = \left\{ \mathbf{W} \in \mathbb{R}^{N \times M} : \mathbf{W} \text{ attains the infimum given (8)} \right\}$. *A weight matrix* $\mathbf{W}$ *is "good" if* $\mathbf{W} \in \mathcal{W}(\mathbf{D})$.

In the above definition, problem (8) is feasible and attainable, i.e., $\mathcal{W}(\mathbf{D}) \neq \varnothing$, which was proven in Lemma 1 of (Chen et al., 2018).

**Theorem 1** (Recovery error upper bound). *Take any* $\mathbf{x}^* \in \mathcal{X}(B,s)$, *any* $\mathbf{W} \in \mathcal{W}(\mathbf{D})$, *and any sequence* $\gamma^{(k)} \in (0, \frac{2}{2\tilde{\mu}s-\tilde{\mu}+1})$. *Using them, define the parameters* $\{\mathbf{W}^{(k)}, \theta^{(k)}\}$:

$$\mathbf{W}^{(k)} = \gamma^{(k)} \mathbf{W}, \quad \theta^{(k)} = \gamma^{(k)} \tilde{\mu}(\mathbf{D}) \sup_{\mathbf{x}^* \in \mathcal{X}(B,s)} \left\{ \|\mathbf{x}^{(k)}(\mathbf{x}^*) - \mathbf{x}^*\|_1 \right\}, \tag{9}$$

*while the sequence* $\{\mathbf{x}^{(k)}(\mathbf{x}^*)\}_{k=1}^{\infty}$ *is generated by (6) using the above parameters and* $\mathbf{x}^{(0)} = \mathbf{0}$ *(Note that each* $\mathbf{x}^{(k)}(\mathbf{x}^*)$ *depends only on* $\theta^{(k-1)}, \theta^{(k-2)}, \ldots$ *and defines* $\theta^{(k)}$*). Let Assumption 1 hold with any* $B > 0$ *and* $s < (1 + 1/\tilde{\mu})/2$. *Then, we have*

$$\text{support}(\mathbf{x}^{(k)}(\mathbf{x}^*)) \subset \mathbb{S}, \quad \|\mathbf{x}^{(k)}(\mathbf{x}^*) - \mathbf{x}^*\|_2 \leq sB \exp\left( -\sum_{\tau=0}^{k-1} c^{(\tau)} \right), \quad k = 1, 2, \ldots \tag{10}$$

*where* $\mathbb{S}$ *is the support of* $\mathbf{x}^*$ *and* $c^{(k)} = -\log\left( (2\tilde{\mu}s - \tilde{\mu})\gamma^{(k)} + |1 - \gamma^{(k)}| \right)$ *is a positive constant.*

In Theorem 1, Eqn. (9) defines the properties of "good" parameters:

- The weights $\mathbf{W}^{(k)}$ can be separated as the product of a scalar $\gamma^{(k)}$ and a matrix $\mathbf{W}$ independent of layer index $k$, where $\mathbf{W}$ has small coherence with $\mathbf{D}$.
- $\gamma^{(k)}$ is bounded in an interval.
- $\theta^{(k)}/\gamma^{(k)}$ is proportional to the $\ell_1$ error of the output of the $k^{\text{th}}$ layer.

The factor $c^{(k)}$ takes the maximum at $\gamma^{(k)} = 1$. If $\gamma^{(k)} \equiv 1$, the recovery error converges to zero in a linear rate (Chen et al., 2018):

$$\|\mathbf{x}^{(k)}(\mathbf{x}^*) - \mathbf{x}^*\|_2 \leq sB \exp\left( -ck \right),$$

where $c = -\log(2\tilde{\mu}s - \tilde{\mu}) \geq c^{(k)}$. Although $\gamma^{(k)} \equiv 1$ gives the optimal theoretical upper bound if there are infinitely many layers $k = 0, 1, 2, \cdots$, it is not the optimal choice for finite $k$. Practically, there are finitely many layers and $\gamma^{(k)}$ obtained by learning is bounded in an interval.

## 2.2 RECOVERY ERROR LOWER BOUND

In this subsection, we introduce a lower bound of the recovery error of LISTA, which illustrates that the parameters analytically given by (9) are optimal in the convergence order (linear).

**Assumption 2.** *The signal* $\mathbf{x}^*$ *is a random variable following the distribution* $P_X$. *Let* $\mathbb{S} = \text{support}(\mathbf{x}^*)$. $P_X$ *satisfies:* $2 \leq |\mathbb{S}| \leq s$; $\mathbb{S}$ *uniformly distributes on the whole index set; non-zero part* $\mathbf{x}_{\mathbb{S}}^*$ *satisfies the uniform distribution with bound* $B$: $|x_i^*| \leq B, \forall i \in \mathbb{S}$. *Moreover, the observation noise* $\varepsilon = \mathbf{0}$.

Theorem 1 tells that an ideal weight $\mathbf{W} \in \mathcal{W}(\mathbf{D})$ satisfies $\mathbf{I} - \mathbf{W}^T\mathbf{D} \approx \mathbf{0}$. But this cannot be met exactly in the overcomplete $\mathbf{D}$ case, i.e., $N < M$. Definition 2 defines the set of matrices $\mathbf{W}$ such that $\mathbf{W}^T\mathbf{D}$ is bounded away from the identity $\mathbf{I}$. In Appendix D, we discuss the feasibility of (11).

**Definition 2.** *Given* $\mathbf{D} \in \mathbb{R}^{N \times M}, s \geq 2, \bar{\sigma}_{\min} > 0$, *we define a set that* $\mathbf{W}^{(k)}$ *are chosen from:*

$$\bar{\mathcal{W}}(\mathbf{D}, s, \bar{\sigma}_{\min}) = \left\{ \mathbf{W} \in \mathbb{R}^{N \times M} \middle| \sigma_{\min}\left( \mathbf{I} - (\mathbf{W}_{:,\mathbb{S}})^T \mathbf{D}_{:,\mathbb{S}} \right) \geq \bar{\sigma}_{\min}, \forall \mathbb{S} \text{ with } 2 \leq |\mathbb{S}| \leq s \right\}. \tag{11}$$

Based on Definition 2, we define a set that $\Theta = \{\mathbf{W}^{(k)}, \theta^{(k)}\}_{k=0}^{\infty}$ are chosen from:

**Definition 3.** *Let $\{\mathbf{x}^{(k)}(\mathbf{x}^*)\}_{k=1}^{\infty}$ be generated by (6) with $\{\mathbf{W}^{(k)}, \theta^{(k)}\}_{k=0}^{\infty}$ and $\mathbf{x}^{(0)} = \mathbf{0}$. Then we define $\mathcal{T}$ as the set of parameters that guarantee there is no false positive in $\mathbf{x}^{(k)}$:*

$$\mathcal{T} = \left\{\{\mathbf{W}^{(k)} \in \bar{\mathcal{W}}(\mathbf{D}, s, \bar{\sigma}_{min}), \theta^{(k)}\}_{k=0}^{\infty} \Big| \mathrm{support}(\mathbf{x}^{(k)}(\mathbf{x}^*)) \subset \mathbb{S}, \ \forall \mathbf{x}^* \in \mathcal{X}(B, s), \ \forall k\right\} \quad (12)$$

The conclusion (10) demonstrates that $\mathcal{T}$ is nonempty because "support$(\mathbf{x}^{(k)}(\mathbf{x}^*)) \subset \mathbb{S}$" is satisfied as long as $\theta^{(k-1)}$ large enough. Actually, $\mathcal{T}$ contains almost all "good" parameters because considerable false positives lead to large recovery errors. With $\mathcal{T}$ defined, we have:

**Theorem 2** (Recovery error lower bound). *Let the sequence $\{\mathbf{x}^{(k)}(\mathbf{x}^*)\}_{k=1}^{\infty}$ be generated by (6) with $\{\mathbf{W}^{(k)}, \theta^{(k)}\}_{k=0}^{\infty}$ and $\mathbf{x}^{(0)} = \mathbf{0}$. Under Assumption 2, for all parameters $\{\mathbf{W}^{(k)}, \theta^{(k)}\}_{k=0}^{\infty} \in \mathcal{T}$ and any sufficient small $\epsilon > 0$, we have*

$$\|\mathbf{x}^{(k)}(\mathbf{x}^*) - \mathbf{x}^*\|_2 \geq \epsilon \|\mathbf{x}^*\|_2 \exp(-\bar{c}k), \quad (13)$$

*with probability at least $(1 - \epsilon s^{3/2} - \epsilon^2)$, where $\bar{c} = s \log(3) - \log(\bar{\sigma}_{min})$.*

This theorem illustrates that, with high probability, the convergence rate of LISTA cannot be faster than a linear rate. Thus, the parameters given in (9), that leads to the linear convergence if $\gamma^k$ is bounded within an interval near 1, are optimal with respect to the order of convergence of LISTA.

## 2.3 ANALYTIC LISTA: LESS PARAMETERS TO LEARN

Following Theorems 1 and 2, we set $\mathbf{W}^{(k)} = \gamma^{(k)}\mathbf{W}$, where $\gamma^{(k)}$ is a scalar, and propose *Tied LISTA (TiLISTA)*:
$$\mathbf{x}^{(k+1)} = \eta_{\theta^{(k)}}\left(\mathbf{x}^{(k)} - \gamma^{(k)}\mathbf{W}^T(\mathbf{D}\mathbf{x}^{(k)} - \mathbf{b})\right), \quad (14)$$

where $\Theta = \left\{\{\gamma^{(k)}\}_k, \{\theta^{(k)}\}_k, \mathbf{W}\right\}$ are parameters to train. The matrix $\mathbf{W}$ is tied over all the layers. Further, we notice that the selection of $\mathbf{W}$ from $\mathcal{W}(\mathbf{D})$ depends on $\mathbf{D}$ only. Hence we propose the **analytic LISTA (ALISTA)** that decomposes tied-LISTA into two stages:

$$\mathbf{x}^{(k+1)} = \eta_{\theta^{(k)}}\left(\mathbf{x}^{(k)} - \gamma^{(k)}\tilde{\mathbf{W}}^T(\mathbf{D}\mathbf{x}^{(k)} - \mathbf{b})\right), \quad (15)$$

where $\tilde{\mathbf{W}}$ is pre-computed by solving the following problem (**Stage 1**)[3]:

$$\tilde{\mathbf{W}} \in \underset{\mathbf{W} \in \mathbb{R}^{N \times M}}{\arg\min} \left\|\mathbf{W}^T\mathbf{D}\right\|_F^2, \quad \text{s.t. } (\mathbf{W}_{:,m})^T\mathbf{D}_{:,m} = 1, \ \forall m = 1, 2, \cdots, M, \quad (16)$$

Then with $\tilde{\mathbf{W}}$ fixed, $\{\gamma^{(k)}, \theta^{(k)}\}_k$ in (15) are learned from end to end (**Stage 2**). (16) reformulates (8) to minimizing the Frobenius norm of $\mathbf{W}^T\mathbf{D}$ (a quadratic objective), over linear constraints. This is a standard convex quadratic program, which is easier to solve than to solve (8) directly.

Table 1: Summary: variants of LISTA and the number of parameters to learn.

| Vanilla LISTA (4) | LISTA-CPSS (6) | TiLISTA (14) | ALISTA (15) |
|---|---|---|---|
| $O(KM^2 + K + MN)$ | $O(KNM + K)$ | $O(NM + K)$ | $O(K)$ |

## 3 CONVOLUTIONAL ANALYTIC LISTA

We extend the analytic LISTA to the convolutional case in this section, starting from discussing the convolutional sparse coding (CSC). Many works studied CSC and proposed efficient algorithms for that (Bristow et al., 2013; Heide et al., 2015; Wohlberg, 2014; 2016; Papyan et al., 2017; Garcia-Cardona & Wohlberg, 2018; Wang et al., 2018; Liu et al., 2017; 2018). In CSC, the general linear transform is replaced by convolutions in order to learn spatially invariant features:

$$\mathbf{b} = \sum_{m=1}^{M} \mathbf{d}_m * \mathbf{x}_m^* + \varepsilon, \quad (17)$$

where each $\mathbf{d}_m$ is a dictionary kernel (or filter). $\{\mathbf{d}_m\}_{m=1}^{M}$ is the dictionary of filters, $M$ denotes the number of filters. $\{\mathbf{x}_m^*\}_{m=1}^{M}$ is the set of coefficient maps that are assumed to have sparse structure,

---

[3]Some details and a complexity analysis of Stage 1 are discussed in Appendix E.1

and $*$ is the convolution operator. Now we consider 2D convolution and take[4] $\mathbf{b} \in \mathbb{R}^{N^2}, \mathbf{d}_m \in \mathbb{R}^{D^2}, \mathbf{x}_m \in \mathbb{R}^{(N+D-1)^2}$. Equation (17) is pointwisely defined as[5]:

$$\mathbf{b}(i,j) = \sum_{k=0}^{D-1} \sum_{l=0}^{D-1} \sum_{m=1}^{M} \mathbf{d}_m(k,l)\mathbf{x}_m(i+k,j+l) + \varepsilon(i,j), \quad 0 \le i,j \le N-1. \quad (18)$$

We concatenate $\mathbf{d}_m$s and $\mathbf{x}_m$s: $\mathbf{d} = [\mathbf{d}_1, \cdots, \mathbf{d}_M]^T$, $\mathbf{x} = [\mathbf{x}_1, \cdots, \mathbf{x}_M]^T$, and rewrite (18) as:

$$\mathbf{b} = \sum_{m=1}^{M} \mathbf{D}_{\text{conv},m}^N(\mathbf{d}_m)\mathbf{x}_m + \varepsilon = \mathbf{D}_{\text{conv}}^N(\mathbf{d})\mathbf{x} + \varepsilon, \quad (19)$$

where the matrix $\mathbf{D}_{\text{conv}}^N(\mathbf{d}) = [\mathbf{D}_{\text{conv},1}^N(\mathbf{d}_1), \cdots, \mathbf{D}_{\text{conv},M}^N(\mathbf{d}_M)] \in \mathbb{R}^{N^2 \times (N+D-1)^2 M}$, depending on the signal size $N$ and the dictionary $\mathbf{d}$, is defined in detail in (48) in Appendix C.2.

From (17), the convolutional LISTA becomes a natural extension of the fully-connected LISTA (6):

$$\mathbf{x}_m^{(k+1)} = \eta_{\theta^{(k)}}\left(\mathbf{x}_m^{(k)} - \left(\mathbf{w}_m^{(k)}\right)' * \left(\sum_{\bar{m}=1}^{M} \mathbf{d}_{\bar{m}} * \mathbf{x}_{\bar{m}}^{(k)} - \mathbf{b}\right)\right), \quad m = 1,2,\cdots,M, \quad (20)$$

where $\{\mathbf{w}_m^{(k)}\}_{m=1}^M$ share the same sizes with $\{\mathbf{d}_m\}_{m=1}^M$ and $(\cdot)'$ means a 180 rotation of the filter (Chalasani et al., 2013). We concatenate the filters together: $\mathbf{w}^{(k)} = [\mathbf{w}_1^{(k)}, \cdots, \mathbf{w}_M^{(k)}]^T \in \mathbb{R}^{D^2 M}$. Parameters to train are $\Theta = \{\mathbf{w}^{(k)}, \theta^{(k)}\}_k$.

Let $\mathbf{W}_{\text{conv}}^N(\mathbf{w}^{(k)})$ be the matrix induced by dictionary $\mathbf{w}^{(k)}$ with the same dimensionality as $\mathbf{D}_{\text{conv}}^N(\mathbf{d})$. Since convolution can be written as a matrix form (19), (20) is equivalent to

$$\mathbf{x}^{(k+1)} = \eta_{\theta^{(k)}}\left(\mathbf{x}^{(k)} - (\mathbf{W}_{\text{conv}}^N(\mathbf{w}^{(k)}))^T(\mathbf{D}_{\text{conv}}^N(\mathbf{d})\mathbf{x}^{(k)} - \mathbf{b})\right). \quad (21)$$

Then by just substituting $\mathbf{D}, \mathbf{W}^{(k)}$ with $\mathbf{D}_{\text{conv}}^N(\mathbf{d}), \mathbf{W}_{\text{conv}}^N(\mathbf{w}^{(k)})$ respectively, Theorems 1 and 2 can be applied to the convolutional LISTA.

**Proposition 1.** *Let $\mathbf{D} = \mathbf{D}_{\text{conv}}^N(\mathbf{d})$ and $\mathbf{W}^{(k)} = \mathbf{W}_{\text{conv}}^N(\mathbf{w}^{(k)})$. With Assumption 1 and other settings the same with those in Theorem 1, (10) holds. With Assumption 2 and other settings the same with those in Theorem 2, (13) holds.*

Similar to the fully connected case (15), based on the results in Proposition 1, we should set $\mathbf{w}_m^{(k)} = \gamma_m^{(k)}\tilde{\mathbf{w}}_m$, $m = 1,2,\cdots,M$, where $\tilde{\mathbf{w}} = [\tilde{\mathbf{w}}_1, \cdots, \tilde{\mathbf{w}}_M]^T$ is chosen from

$$\tilde{\mathbf{w}} \in \mathcal{W}_{\text{conv}}^N = \underset{\substack{\mathbf{w} \in \mathbb{R}^{D^2 M} \\ \mathbf{w}_m \cdot \mathbf{d}_m = 1, \, 1 \le m \le M}}{\arg\min} \left\|\left(\mathbf{W}_{\text{conv}}^N(\mathbf{w})\right)^T \mathbf{D}_{\text{conv}}^N(\mathbf{d})\right\|_F^2. \quad (22)$$

However, (22) is not as efficient to solve as (16). To see that, matrices $\mathbf{D}_{\text{conv}}^N(\mathbf{d})$ and $\mathbf{W}_{\text{conv}}^N(\mathbf{w})$ are both of size $N^2 \times (N+D-1)^2 M$, the coherence matrix $\left(\mathbf{W}_{\text{conv}}^N(\mathbf{w})\right)^T \mathbf{D}_{\text{conv}}^N(\mathbf{d})$ is thus of size $(N+D-1)^2 M \times (N+D-1)^2 M$. In the typical application setting of CSC, $\mathbf{b}$ is usually an image rather than a small patch. For example, if the image size is $100 \times 100$, dictionary size is $7 \times 7 \times 64$, $N = 100, D = 7, M = 64$, then $(N+D-1)^2 M \times (N+D-1)^2 M \approx 5 \times 10^{11}$.

## 3.1 CALCULATING CONVOLUTIONAL WEIGHTS ANALYTICALLY AND EFFICIENTLY

To overcome the computational challenge of solving (22), we exploit the following *circular convolution* as an efficient approximation:

$$\mathbf{b}(i,j) = \sum_{k=0}^{D-1} \sum_{l=0}^{D-1} \sum_{m=1}^{M} \mathbf{d}_m(k,l)\mathbf{x}_m\left((i+k)_{\text{mod}N}, (j+l)_{\text{mod}N}\right) + \varepsilon(i,j), \quad 0 \le i,j \le N-1, \quad (23)$$

---

[4]Here, $\mathbf{b}, \mathbf{d}_m, \mathbf{x}_m$ are vectors. The notion $\mathbf{b}(i,j)$ means the $(iN+j)^{\text{th}}$ entry of $\mathbf{b}$. Additionally, $\mathbf{d}_m, \mathbf{x}_m$ are defined in the same way for all $m = 1, \cdots, M$.

[5]Strictly speaking, (18) is the cross-correlation rather than convolution. However in TensorFlow, that operation is named as convolution, and we follow that convention to be consistent with the learning community.

where $\mathbf{b} \in \mathbb{R}^{N^2}, \mathbf{d}_m \in \mathbb{R}^{D^2}, \mathbf{x}_m \in \mathbb{R}^{N^2}$. Similar to (18), we rewrite (23) in a compact way:

$$\mathbf{b} = \sum_{m=1}^{M} \mathbf{D}_{\mathrm{cir},m}^{N}(\mathbf{d}_m)\mathbf{x}_m + \varepsilon = \mathbf{D}_{\mathrm{cir}}^{N}(\mathbf{d})\mathbf{x} + \varepsilon,$$

where $\mathbf{D}_{\mathrm{cir}}^{N}(\mathbf{d}) : \mathbb{R}^{N^2 M} \to \mathbb{R}^{N^2}$ is a matrix depending on the signal size $N$ and the dictionary $\mathbf{d}$. Then the coherence minimization with the circular convolution is given by

$$\mathcal{W}_{\mathrm{cir}}^{N} = \underset{\substack{\mathbf{w} \in \mathbb{R}^{D^2 M} \\ \mathbf{w}_m \cdot \mathbf{d}_m = 1, \ 1 \leq m \leq M}}{\arg\min} \left\| \left(\mathbf{W}_{\mathrm{cir}}^{N}(\mathbf{w})\right)^T \mathbf{D}_{\mathrm{cir}}^{N}(\mathbf{d}) \right\|_F^2. \tag{24}$$

The following theorem motivates us to use the solution to (24) to approximate that of (22).

**Theorem 3.** *The solution sets of (22) and (24) satisfy the following properties:*

1. *$\mathcal{W}_{\mathrm{cir}}^{N} = \mathcal{W}_{\mathrm{cir}}^{2D-1}, \forall N \geq 2D - 1$.*

2. *If at least one of the matrices $\{\mathbf{D}_{\mathrm{cir},1}^{2D-1}, \cdots, \mathbf{D}_{\mathrm{cir},M}^{2D-1}\}$ is non-singular, $\mathcal{W}_{\mathrm{cir}}^{2D-1}$ involves only a unique element. Furthermore,*

$$\lim_{N \to \infty} \mathcal{W}_{\mathrm{conv}}^{N} = \mathcal{W}_{\mathrm{cir}}^{2D-1}. \tag{25}$$

The solution set $\mathcal{W}_{\mathrm{cir}}^{N}$ is not related with the image size $N$ as long as $N \geq 2D - 1$, thus one can deal with a much smaller-size problem (let $N = 2D - 1$). Further, (25) indicates that as $N$ gets (much) larger than $D$, the boundary condition becomes less important. Thus, one can use $\mathcal{W}_{\mathrm{cir}}^{2D-1}$ to approximate $\mathcal{W}_{\mathrm{conv}}^{N}$. In Appendix E.2, we introduce the algorithm details of solving (24).

Based on Proposition 1 and Theorem 3, we obtain the **convolutional ALISTA**:

$$\mathbf{x}_m^{(k+1)} = \eta_{\theta^{(k)}} \left( \mathbf{x}_m^{(k)} - \gamma_m^{(k)} \left(\tilde{\mathbf{w}}_m\right)' * \left( \sum_{\bar{m}=1}^{M} \mathbf{d}_{\bar{m}} * \mathbf{x}_{\bar{m}}^{(k)} - \mathbf{b} \right) \right), \quad m = 1, 2, \cdots, M, \tag{26}$$

where $\tilde{\mathbf{w}} = [\tilde{\mathbf{w}}_1, \cdots, \tilde{\mathbf{w}}_M]^T \in \mathcal{W}_{\mathrm{cir}}^{2D-1}$ and $\Theta = \{\{\gamma_m^{(k)}\}_{m,k}, \{\theta^{(k)}\}_k\}$ are the parameters to train. (26) is a simplified form, compared to the empirically unfolded CSC model recently proposed in (Sreter & Giryes, 2018)

## 4 ROBUST ALISTA TO MODEL PERTURBATION

Many applications, such as often found in surveillance video scenarios (Zhao et al., 2011; Han et al., 2013), can be formulated as sparse coding models whose dictionaries are subject to small dynamic perturbations (e.g, slowly varied over time). Specifically, the linear system model (1) may have uncertain $\mathbf{D}$: $\tilde{\mathbf{D}} = \mathbf{D} + \varepsilon_{\mathbf{D}}$, where $\varepsilon_{\mathbf{D}}$ is some small stochastic perturbation. Classical LISTA entangles the learning of all its parameters, and the trained model is tied to one static $\mathbf{D}$. The important contribution of ALISTA is to decompose fitting $\mathbf{W}$ w.r.t. $\mathbf{D}$, from adapting other parameters $\{\gamma^{(k)}, \theta^{(k)}\}_k$ to training data.

In this section, we develop a robust variant of ALISTA that is a fast regressor not only for a given $\mathbf{D}$, but all its randomly perturbations $\tilde{\mathbf{D}}$ to some extent. Up to our best knowledge, this approach is new. Robust ALISTA can be sketched as the following empirical routine (at each iteration):

- Sample a perturbed dictionary $\tilde{\mathbf{D}}$. Sample $\mathbf{x}$ and $\varepsilon$ to generate $\mathbf{b}$ w.r.t. $\tilde{\mathbf{D}}$.
- Apply Stage 1 of ALISTA w.r.t. $\tilde{\mathbf{D}}$ and obtain $\tilde{\mathbf{W}}$; however, instead of an iterative minimization algorithm, we use a neural network that unfolds that algorithm to produce $\tilde{\mathbf{W}}$.
- Apply Stage 2 of ALISTA w.r.t. $\tilde{\mathbf{W}}, \mathbf{D}, \mathbf{x}$, and $\mathbf{b}$ to obtain $\{\gamma^{(k)}, \theta^{(k)}\}_k$.

In Robust ALISTA above, $\tilde{\mathbf{D}}$ becomes a part of the data for training the neural network that generates $\tilde{\mathbf{W}}$. This neural network is faster to apply than the minimization algorithm. One might attempt to use $\tilde{\mathbf{D}}$ in the last step, rather than $\mathbf{D}$, but $\tilde{\mathbf{D}}$ makes training less stable, potentially because of larger weight variations between training iterations due to the random perturbations in $\tilde{\mathbf{D}}$. We observe that using $\mathbf{D}$ stabilizes training better and empirically achieves a good prediction. More details of training Robust ALISTA are given in Appendix G.

# 5 NUMERICAL RESULTS

In this section, we conduct extensive experiments on both synthesized and real data to demonstrate:[6]

- We experimentally validate Theorems 1 and 2, and show that ALISTA is as effective as classical LISTA (Gregor & LeCun, 2010; Chen et al., 2018)but is much easier to train.
- Similar conclusions can be drawn for convolutional analytic LISTA.
- The robust analytic LISTA further shows remarkable robustness in sparse code prediction, given that $\mathbf{D}$ is randomly perturbed within some extent.

**Notation** For brevity, we let *LISTA* denote the vanilla LISTA model (4) in (Gregor & LeCun, 2010); *LISTA-CPSS* refers to the lately-proposed fast LISTA variant (Chen et al., 2018) with weight coupling and support selection; *TiLISTA* is the tied LISTA (14); and *ALISTA* is our proposed Analytic LISTA (15). If the model is for convolutional case, then we add "*Conv*" as the prefix for model name, such as "*Conv ALISTA*" that represents the convolutional analytic LISTA.

## 5.1 VALIDATION OF THEOREMS 1 AND 2 (ANALYTIC LISTA)

We follow the same $N = 250, M = 500$ setting as (Chen et al., 2018) by default. We sample the entries of $\mathbf{D}$ i.i.d. from the standard Gaussian distribution, $\mathbf{D}_{ij} \sim \mathcal{N}(0, 1/N)$ and then normalize its columns to have the unit $\ell_2$ norm. We fix a dictionary $\mathbf{D}$ in this section. To generate sparse vectors $\mathbf{x}^*$, we decide each of its entry to be non-zero following the Bernoulli distribution with $p_b = 0.1$. The values of the non-zero entries are sampled from the standard Gaussian distribution. A test set of 1000 samples generated in the above manner is fixed for all tests in our simulations. The analytic weight $W$ that we use in the ALISTA is obtained by solving (16).

All networks used (vanilla LISTA, LISTA-CPSS, TiLISTA and ALISTA) have the same number of 16 layers. We also include two classical iterative solvers: ISTA and FISTA. We train the networks with four different levels of noises: SNR (Signal-to-Noise Ratio) = $20, 30, 40$, and $\infty$. While our theory mainly discussed the noise-free case (SNR = $\infty$), we hope to empirically study the algorithm performance under noise too. As shown in Figure 1, the x-axes denotes the indices of layers for the networks, or the number of iterations for the iterative algorithms. The y-axes represent the NMSE (Normalized Mean Squared Error) in the decibel (dB) unit:

$$\text{NMSE}_{\text{dB}}(\hat{\mathbf{x}}, \mathbf{x}^*) = 10 \log_{10} \left( \mathbb{E}\|\hat{\mathbf{x}} - \mathbf{x}^*\|^2 / \mathbb{E}\|\mathbf{x}^*\|^2 \right),$$

where $\mathbf{x}^*$ is the ground truth and $\hat{\mathbf{x}}$ is the estimated one.

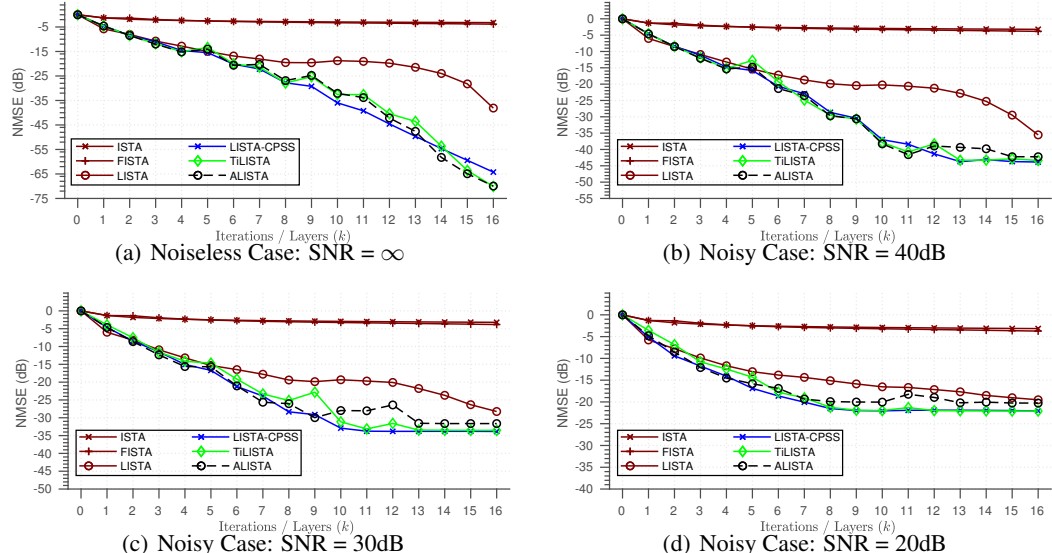

Figure 1: Justification of Theorems 1 and 2: comparision among LISTA variants.

---

[6]Our codes are uploaded to `https://github.com/xchen-tamu/alista`.

In Figure 1 (a) noise-less case, all four learned models apparently converge much faster than two iterative solvers (ISTA/FISTA curves almost overlap in this y-scale, at the small number of iterations). Among the four networks, classical-LISTA is inferior to the other three by an obvious margin. LISTA-CPSS, TiLISTA and ALISTA perform comparably: ALISTA is observed to eventually achieve the lowest NMSE. Figure 1(a) also supports Theorem 2, that all networks have at most linear convergence, regardless of how freely their parameters can be end-to-end learned.

Figure 1 (b) - (d) further show that even in the presence of noise, ALISTA can empirically perform comparably with LISTA-CPSS and TiLISTA, and stay clearly better than LISTA and ISTA/FISTA. Always note that ALISTA the smallest amount of parameters to learn from the end-to-end training (Stage 2). The above results endorse that: i) the optimal LISTA layer-wise weights could be structured as $\mathbf{W}^{(k)} = \gamma^{(k)}\mathbf{W}$; and ii) $\mathbf{W}$ could be analytically solved rather than learned from data, without incurring performance loss. We also observe the significant reduction of training time for ALISTA: while LISTA-CPSS of the same depth took $\sim$1.5 hours to train, ALISTA was trained within only 6 minutes (0.1 hours) to achieve comparable performance, on the same hardware (one 1080 Ti on server).

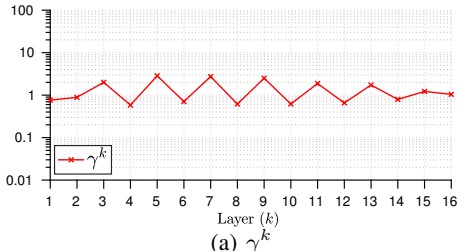 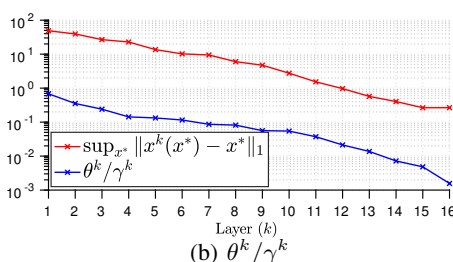

(a) $\gamma^k$      (b) $\theta^k/\gamma^k$

Figure 2: Justification of Theorem 1 (noiseless case): parameters obtained by training satisfy (9).

We further supply Figures 2 and 3 to justify Theorem 1 from different perspectives. Figure 2 plots the learned parameters $\{\gamma^{(k)}, \theta^{(k)}\}$ in ALISTA (Stage 2), showing that they satisfy the properties proposed in Theorem 1: $\gamma^{(k)}$ bounded; $\theta^{(k)}$ and $\gamma^{(k)}$ is proportional to $\sup_{\mathbf{x}^*} \|\mathbf{x}^{(k)}(\mathbf{x}^*) - \mathbf{x}^*\|_1$ ("$\sup_{\mathbf{x}^*}$" is taken over the test set). Figure 3 reports the average magnitude[7] of the false positives and the true positives in $\mathbf{x}^k(\mathbf{x}^*)$ of ALISTA: the "true positives" curve draws the values of $\mathbb{E}\{\|\mathbf{x}^k_{\mathbb{S}}(\mathbf{x}^*)\|_2^2/\|\mathbf{x}^k(\mathbf{x}^*)\|_2^2\}$ w.r.t. $k$ (the expectation is taken over the test set), while "false positives" for

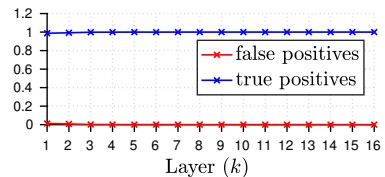

Figure 3: Justification of Theorem 1 (noiseless case): Proportion of false positives vs true positives in $\mathbf{x}^k(\mathbf{x}^*)$.

$\mathbb{E}\{\|\mathbf{x}^k_{\mathbb{S}^c}(\mathbf{x}^*)\|_2^2/\|\mathbf{x}^k(\mathbf{x}^*)\|_2^2\}$. False positives take up small proportion over the positives, which supports the Theorem 1 conclusion that $\text{support}(\mathbf{x}^k(\mathbf{x}^*)) \subset \mathbb{S}$.

### 5.2 VALIDATION OF THEOREM 3 (CONVOLUTIONAL ANALYTIC LISTA)

For convolutional cases, we use real image data to verify Theorem 3. We train a convolutional dictionary $\mathbf{d}$ with $D = 7, M = 64$ on the BSD500 training set (400 images), using the Algorithm 1 in (Liu et al., 2018). We then use it for problems (22) and (24) and solve them with different $N$s.

In Table 2, we take $\mathbf{w}^N_{\text{cir}} \in \mathcal{W}^N_{\text{cir}}$, $\mathbf{w}^* \in \mathcal{W}^{50}_{\text{cir}}$ (consider 50 as large enough) For this example, $\mathcal{W}^N_{\text{cir}}$ has only one element. Table 2 shows that $\mathbf{w}^N_{\text{cir}} = \mathbf{w}^*$ for $N \geq 13$, i.e., the solution of the problem (24) is independent of $N$ if $N \geq 2D - 1$, justifying the first conclusion in Theorem 3. In Table 3, we take $\mathbf{w}^N_{\text{conv}} \in \mathcal{W}^N_{\text{conv}}$ and $\mathbf{w}^* \in \mathbf{w}^{13}_{\text{cir}}$, where $\mathcal{W}^N_{\text{conv}}$ also has only one element. Table 3 shows $\mathbf{w}^N_{\text{conv}} \rightarrow \mathbf{w}^*$, i.e., the solution of the problem (22) converges to that of (24) as $N$ increases, validating the second conclusion of Theorem 3. Visualized $\mathbf{w}^* \in \mathbf{w}^{13}_{\text{cir}}$ is displayed in Appendix F.

---

[7]The number and proportion of false alarms are a more straightforward performance metric. However, they are sensitive to the threshold. We found that, although using a smaller threshold leads to more false alarms, the final recovery quality is better and those false alarms have small magnitudes and are easy to remove by thresholding during post-processing. That's why we chose to show their magnitudes, implying that we get easy-to-remove false alarms.

Besides validating Theorem 3, we also present a real image denoising experiment to verify the effectiveness of Conv ALISTA. The detailed settings and results are presented in Appendix H.

Table 2: Validation of Conclusion 1 in Theorem 3. $D = 7$. $\mathbf{w}_{\text{cir}}^N \in \mathcal{W}_{\text{cir}}^N$ and $\mathbf{w}^* \in \mathcal{W}_{\text{cir}}^{50}$.

| $\|\mathbf{w}_{\text{cir}}^N - \mathbf{w}^*\|^2 / \|\mathbf{w}^*\|^2$ | | | | | |
|---|---|---|---|---|---|
| $N = 10$ | $N = 11$ | $N = 12$ | $N = 13$ | $N = 15$ | $N = 20$ |
| $2.0 \times 10^{-2}$ | $9.3 \times 10^{-3}$ | $3.9 \times 10^{-3}$ | $1.4 \times 10^{-12}$ | $8.8 \times 10^{-13}$ | $5.9 \times 10^{-13}$ |

Table 3: Validation of Conclusion 2 in Theorem 3. $D = 7$. $\mathbf{w}_{\text{conv}}^N \in \mathcal{W}_{\text{conv}}^N$ and $\mathbf{w}^* \in \mathbf{w}_{\text{cir}}^{13}$.

| $\|\mathbf{w}_{\text{conv}}^N - \mathbf{w}^*\|^2 / \|\mathbf{w}^*\|^2$ | | | | |
|---|---|---|---|---|
| $N = 3$ | $N = 5$ | $N = 10$ | $N = 15$ | $N = 20$ |
| 0.1892 | 0.0850 | 0.0284 | 0.0161 | 0.0113 |

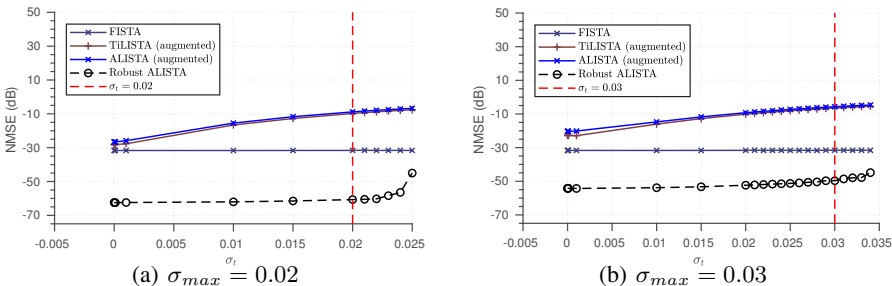

Figure 4: Validation of Robust ALISTA.

## 5.3 VALIDATION OF ROBUST ALISTA

We empirically verify the effectiveness of Robust ALISTA, by sampling the dictionary perturbation $\boldsymbol{\varepsilon}_{\boldsymbol{D}}$ entry-wise i.i.d. from another Gaussian distribution $\mathcal{N}(0, \sigma_{max}^2)$. We choose $\sigma_{max} = 0.02$ and 0.03. Other simulation settings are by default the same as in Section 5.1. We then build the Robust ALISTA model, following the strategy in Section 4 and using a 4-layer encoder for approximating its second step (see Appendix G for details). Correspondingly, we compare Robust ALISTA with TiLISTA and ALISTA with *specific data augmentation*: we straightforwardly augment their training sets, by including all data generated with randomly perturbed $\tilde{\mathbf{D}}$s when training Robust ALISTA. We also include the data-free FISTA algorithm into the comparison.

Figure 4 plots the results when the trained models are applied on the testing data, generated with the same dictionary and perturbed by $\mathcal{N}(0, \sigma_t)$. We vary $\sigma_t$ from zero to slightly above $\sigma_{max}$. Not surprisingly, FISTA is unaffected, while the other three data-driven models all slight degrade as $\sigma_t$ increases. Compared to the augmented TiLISTA and ALISTA whose performance are both inferior to FISTA, the proposed Robust ALISTA appears to be much more favorable in improving robustness to model perturbations. In both $\sigma_{max}$ cases, it consistently achieves much lower NMSE than FISTA, even when $\sigma_t$ has slightly surpassed $\sigma_{max}$. Although the NMSE of ALISTA may decrease faster if $\sigma_t$ continues growing larger, such decrease could be alleviated by improving $\sigma_{max}$ in training, e.g., by comparing $\sigma_{max} = 0.02$ and 0.03. Robust ALISTA demonstrates remarkable robustness and maintains the best NMSE performance, within at least the $[0, \sigma_{max}]$ range.

## 6 CONCLUSIONS AND FUTURE WORK

Based on the recent theoretical advances of LISTA, we have made further steps to reduce the training complexity and improve the robustness of LISTA. Specifically, we no longer train any matrix for LISTA but directly use the solution to an analytic minimization problem to solve for its layer-wise weights. Therefore, only two scalar sequences (stepsizes and thresholds) still need to be trained. Excluding the matrix from training is backed by our theoretical upper and lower bounds. The resulting method, Analytic LISTA or ALISTA, is not only faster to train but performs as well as the state-of-the-art variant of LISTA by (Chen et al., 2018). This discovery motivates us to further replace the minimization algorithm by its unfolding neural network, and train this neural network to more quickly produce the weight matrix. The resulting algorithm is used to handle perturbations in the model dictionary — we only train once for a dictionary with all its small perturbations. Our future work will investigate the theoretical sensitivity of ALISTA (and its convolutional version) to noisy measurements.

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

## A    PROOF OF THEOREM 1

In this proof, we use the notion $\mathbf{x}^{(k)}$ to replace $\mathbf{x}^{(k)}(\mathbf{x}^*)$ for simplicity. We fix $\mathbf{D}$ in the proof, $\tilde{\mu}(\mathbf{D})$ can be simply written as $\tilde{\mu}$.

Before proving Theorem 1, we present and prove a lemma.

**Lemma 1.** *With all the settings the same with those in Theorem 1, we have*

$$\text{support}(\mathbf{x}^{(k)}) \subset \mathbb{S}, \quad \forall k. \tag{27}$$

*In another word, there are no false positives in $\mathbf{x}^{(k)}$: $x_i^{(k)} = 0, \forall i \notin \mathbb{S}, \forall k$.*

*Proof.* Take arbitrary $\mathbf{x}^* \in \mathcal{X}(B, s)$. We prove Lemma 1 by induction. As $k = 0$, (27) is satisfied since $\mathbf{x}^{(0)} = \mathbf{0}$. Fixing $k$, and assuming $\text{support}(\mathbf{x}^{(k)}) \subset \mathbb{S}$, we have

$$
\begin{aligned}
x_i^{(k+1)} &= \eta_{\theta^{(k)}}\left( x_i^{(k)} - \gamma^{(k)}(\mathbf{W}_{:,i})^T(\mathbf{D}\mathbf{x}^{(k)} - \mathbf{b}) \right) \\
&= \eta_{\theta^{(k)}}\left( -\gamma^{(k)}\sum_{j \in \mathbb{S}}(\mathbf{W}_{:,i})^T\mathbf{D}_{:,j}(x_j^{(k)} - x_j^*) \right), \quad \forall i \notin \mathbb{S}.
\end{aligned}
$$

By (9), the thresholds are taken as $\theta^{(k)} = \tilde{\mu}\gamma^{(k)}\sup_{\mathbf{x}^*}\{\|\mathbf{x}^{(k)} - \mathbf{x}^*\|_1\}$. Also, since $\mathbf{W} \in \mathcal{W}(\mathbf{D})$, we have $|(\mathbf{W}_{:,i})^T\mathbf{D}_{:,j}| \leq \tilde{\mu}$ for all $j \neq i$. Thus, for all $i \notin \mathbb{S}$,

$$
\begin{aligned}
\theta^{(k)} &\geq \tilde{\mu}\gamma^{(k)}\|\mathbf{x}^{(k)} - \mathbf{x}^*\|_1 = \sum_{j \in \text{support}(\mathbf{x}^{(k)})}\tilde{\mu}\gamma^{(k)}|x_j^{(k)} - x_j^*| = \sum_{j \in \mathbb{S}}\tilde{\mu}\gamma^{(k)}|x_j^{(k)} - x_j^*| \\
&\geq \left| -\gamma^{(k)}\sum_{j \in \mathbb{S}}(\mathbf{W}_{:,i})^T\mathbf{D}_{:,j}(x_j^{(k)} - x_j^*) \right|,
\end{aligned}
$$

which implies $x_i^{(k+1)} = 0, \forall i \notin \mathbb{S}$ by the definition of $\eta_{\theta^{(k)}}$, i.e.,

$$\text{support}(\mathbf{x}^{(k+1)}) \subset \mathbb{S}$$

By induction, (27) is proved. $\qquad\square$

With Lemma 1, we are able to prove Theorem 1 now.

*Proof of Theorem 1.* Take arbitrary $\mathbf{x}^* \in \mathcal{X}(B, s)$. For all $i \in \mathbb{S}$, by (27), we obtain

$$
\begin{aligned}
x_i^{(k+1)} &= \eta_{\theta^{(k)}}\left( x_i^{(k)} - \gamma^{(k)}(\mathbf{W}_{:,i})^T\mathbf{D}_{:,\mathbb{S}}(\mathbf{x}_{\mathbb{S}}^{(k)} - \mathbf{x}_{\mathbb{S}}^*) \right) \\
&\in x_i^{(k)} - \gamma^{(k)}(\mathbf{W}_{:,i})^T\mathbf{D}_{:,\mathbb{S}}(\mathbf{x}_{\mathbb{S}}^{(k)} - \mathbf{x}_{\mathbb{S}}^*) - \theta^{(k)}\partial\ell_1(x_i^{(k+1)}),
\end{aligned}
$$

where $\partial\ell_1(x)$ is the sub-gradient of $|x|, x \in \mathbb{R}$:

$$
\partial\ell_1(x) = \begin{cases} \{\text{sign}(x)\} & \text{if } x \neq 0, \\ [-1, 1] & \text{if } x = 0. \end{cases}
$$

The choice of $\mathbf{W} \in \mathcal{W}(\mathbf{D})$ gives $(\mathbf{W}_{:,i})^T\mathbf{D}_{:,i} = 1$. Thus,

$$
\begin{aligned}
&x_i^{(k)} - \gamma^{(k)}(\mathbf{W}_{:,i})^T\mathbf{D}_{:,\mathbb{S}}(\mathbf{x}_{\mathbb{S}}^{(k)} - \mathbf{x}_{\mathbb{S}}^*) \\
=&x_i^{(k)} - \gamma^{(k)}\sum_{j \in \mathbb{S}, j \neq i}(\mathbf{W}_{:,i})^T\mathbf{D}_{:,j}(x_j^{(k)} - x_j^*) - \gamma^{(k)}(x_i^{(k)} - x_i^*) \\
=&x_i^* - \gamma^{(k)}\sum_{j \in \mathbb{S}, j \neq i}(\mathbf{W}_{:,i})^T\mathbf{D}_{:,j}(x_j^{(k)} - x_j^*) + (1 - \gamma^{(k)})(x_i^{(k)} - x_i^*).
\end{aligned}
$$

Then the following inclusion formula holds for all $i \in \mathbb{S}$,

$$x_i^{(k+1)} - x_i^* \in -\gamma^{(k)}\sum_{j \in \mathbb{S}, j \neq i}(\mathbf{W}_{:,i})^T\mathbf{D}_{:,j}(x_j^{(k)} - x_j^*) - \theta^{(k)}\partial\ell_1(x_i^{(k+1)}) + (1 - \gamma^{(k)})(x_i^{(k)} - x_i^*).$$

By the definition of $\partial \ell_1$, every element in $\partial \ell_1(x), \forall x \in \mathbb{R}$ has a magnitude less than or equal to 1. Thus, for all $i \in \mathbb{S}$,

$$|x_i^{(k+1)} - x_i^*| \leq \sum_{j \in \mathbb{S}, j \neq i} \gamma^{(k)} \left| (\mathbf{W}_{:,i})^T \mathbf{D}_{:,j} \right| |x_j^{(k)} - x_j^*| + \theta^{(k)} + |1 - \gamma^{(k)}| |x_i^{(k)} - x_i^*|$$

$$\leq \tilde{\mu} \gamma^{(k)} \sum_{j \in \mathbb{S}, j \neq i} |x_j^{(k)} - x_j^*| + \theta^{(k)} + |1 - \gamma^{(k)}| |x_i^{(k)} - x_i^*|.$$

Equation (27) implies $\|\mathbf{x}^{(k)} - \mathbf{x}^*\|_1 = \|\mathbf{x}_{\mathbb{S}}^{(k)} - \mathbf{x}_{\mathbb{S}}^*\|_1$ for all $k$. Then

$$\|\mathbf{x}^{(k+1)} - \mathbf{x}^*\|_1 = \sum_{i \in \mathbb{S}} |x_i^{(k+1)} - x_i^*|$$

$$\leq \sum_{i \in \mathbb{S}} \left( \tilde{\mu} \gamma^{(k)} \sum_{j \in \mathbb{S}, j \neq i} |x_j^{(k)} - x_j^*| + \theta^{(k)} + |1 - \gamma^{(k)}| |x_i^{(k)} - x_i^*| \right)$$

$$= \tilde{\mu} \gamma^{(k)} (|\mathbb{S}| - 1) \sum_{i \in \mathbb{S}} |x_i^{(k)} - x_i^*| + \theta^{(k)} |\mathbb{S}| + |1 - \gamma^{(k)}| \|\mathbf{x}^{(k)} - \mathbf{x}^*\|_1$$

$$= \tilde{\mu} \gamma^{(k)} (|\mathbb{S}| - 1) \|\mathbf{x}^{(k)} - \mathbf{x}^*\|_1 + \theta^{(k)} |\mathbb{S}| + |1 - \gamma^{(k)}| \|\mathbf{x}^{(k)} - \mathbf{x}^*\|_1.$$

Taking supremum of the above inequality over $\mathbf{x}^* \in \mathcal{X}(B, s)$, by $|\mathbb{S}| \leq s$,

$$\sup_{\mathbf{x}^*} \{\|\mathbf{x}^{(k+1)} - \mathbf{x}^*\|_1\} \leq \left( \tilde{\mu} \gamma^{(k)} (s - 1) + |1 - \gamma^{(k)}| \right) \sup_{\mathbf{x}^*} \{\|\mathbf{x}^{(k)} - \mathbf{x}^*\|_1\} + \theta^{(k)} s.$$

By the value of $\theta^{(k)}$ given in (9), we have

$$\sup_{\mathbf{x}^*} \{\|\mathbf{x}^{(k+1)} - \mathbf{x}^*\|_1\} \leq \left( \gamma^{(k)} (2\tilde{\mu} s - \tilde{\mu}) + |1 - \gamma^{(k)}| \right) \sup_{\mathbf{x}^*} \{\|\mathbf{x}^{(k)} - \mathbf{x}^*\|_1\}.$$

Let $c^{(\tau)} = -\log \left( (2\tilde{\mu} s - \tilde{\mu}) \gamma^{(\tau)} + |1 - \gamma^{(\tau)}| \right)$. Then, by induction,

$$\sup_{\mathbf{x}^*} \{\|\mathbf{x}^{(k+1)} - \mathbf{x}^*\|_1\} \leq \exp\left( -\sum_{\tau=0}^{k} c^{(\tau)} \right) \sup_{\mathbf{x}^*} \{\|\mathbf{x}^{(0)} - \mathbf{x}^*\|_1\} \leq \exp\left( -\sum_{\tau=0}^{k} c^{(\tau)} \right) sB.$$

Since $\|\mathbf{x}\|_2 \leq \|\mathbf{x}\|_1$ for any $\mathbf{x} \in \mathbb{R}^n$, we can get the upper bound for $\ell_2$ norm:

$$\sup_{\mathbf{x}^*} \{\|\mathbf{x}^{(k+1)} - \mathbf{x}^*\|_2\} \leq \sup_{\mathbf{x}^*} \{\|\mathbf{x}^{(k+1)} - \mathbf{x}^*\|_1\} \leq sB \exp\left( -\sum_{\tau=0}^{k} c^{(\tau)} \right).$$

The assumption $s < (1 + 1/\tilde{\mu})/2$ gives $2\tilde{\mu} s - \tilde{\mu} < 1$. If $0 < \gamma^{(k)} \leq 1$, we have $c^{(k)} > 0$. If $1 < \gamma^{(k)} < 2/(1 + 2\tilde{\mu} s - \tilde{\mu})$, we have

$$(2\tilde{\mu} s - \tilde{\mu}) \gamma^{(k)} + |1 - \gamma^{(k)}| = (2\tilde{\mu} s - \tilde{\mu}) \gamma^{(k)} + \gamma^{(k)} - 1 < 1,$$

which implies $c^{(k)} > 0$. Theorem 1 is proved. $\qquad \square$

## B  PROOF OF THEOREM 2

*Proof of Theorem 2.* We fix $\mathbf{D}$ and sample a $\mathbf{x}^* \sim P_X$.

If we can prove

$$P\left( (13) \text{ does not hold} \,\middle|\, \text{support}(\mathbf{x}^*) = \mathbb{S} \right) \leq \epsilon |\mathbb{S}| + \epsilon^{|\mathbb{S}|}, \tag{28}$$

then the lower bound (13) in Theorem 2 is proved by

$$P\left( (13) \text{ holds} \right) = \sum_{\mathbb{S}, 2 \leq |\mathbb{S}| \leq s} P\left( (13) \text{ holds} \,\middle|\, \text{support}(\mathbf{x}^*) = \mathbb{S} \right) P\left( \text{support}(\mathbf{x}^*) = \mathbb{S} \right)$$

$$\geq (1 - \epsilon s^{3/2} - \epsilon^2) \sum_{2 \leq |\mathbb{S}| \leq s} P\left( \text{support}(\mathbf{x}^*) = \mathbb{S} \right)$$

$$= 1 - \epsilon s^{3/2} - \epsilon^2.$$

Now we fix $k$ and prove inequality (28) by three steps:

**Step 1: If (13) does not hold, then what condition $\mathbf{x}^*$ should satisfy?**

Fixing $k$, we define a set $\mathcal{X}^{(k)}(\epsilon)$, which involves all the $\mathbf{x}^*$ that does not satisfy (13):

$$\mathcal{X}^{(k)}(\epsilon) = \{(13) \text{ does not hold}\} = \left\{\mathbf{x}^* \Big| \|\mathbf{x}^{(k)}(\mathbf{x}^*) - \mathbf{x}^*\|_2 < \epsilon \|\mathbf{x}^*\|_2 \Big(\frac{\bar{\sigma}_{\min}}{3^s}\Big)^k\right\}.$$

Let $\mathbb{S} = \text{support}(\mathbf{x}^*)$. For $\mathbf{x}^* \in \mathcal{X}^{(k)}(\epsilon)$, we consider two cases:

1. $|x_i^*| > \epsilon \|\mathbf{x}^*\|_2 (\bar{\sigma}_{\min}/3^s)^k$, $\forall i \in \mathbb{S}$.

2. $|x_i^*| \leq \epsilon \|\mathbf{x}^*\|_2 (\bar{\sigma}_{\min}/3^s)^k$, for some $i \in \mathbb{S}$.

If case 1 holds, we obtain that the support of $\mathbf{x}^{(k)}$ is exactly the same with that of $\mathbf{x}^*$:

$$\text{support}(\mathbf{x}^{(k)}(\mathbf{x}^*)) = \mathbb{S}.$$

Then the relationship between $\mathbf{x}^{(k)}$ and $\mathbf{x}^{(k-1)}$ can be reduced to an affine transform:

$$\begin{aligned}
\mathbf{x}_{\mathbb{S}}^{(k)} &= \eta_{\theta^{(k)}}\Big(\mathbf{x}_{\mathbb{S}}^{(k-1)} - (\mathbf{W}_{:,\mathbb{S}}^{(k-1)})^T(\mathbf{D}\mathbf{x}^{(k-1)} - \mathbf{b})\Big) \\
&= \mathbf{x}_{\mathbb{S}}^{(k-1)} - (\mathbf{W}_{:,\mathbb{S}}^{(k-1)})^T\mathbf{D}_{:,\mathbb{S}}(\mathbf{x}_{\mathbb{S}}^{(k-1)} - \mathbf{x}_{\mathbb{S}}^*) - \theta^{(k-1)}\text{sign}(\mathbf{x}_{\mathbb{S}}^{(k)}).
\end{aligned} \tag{29}$$

Subtracting $\mathbf{x}^*$ from the two sides of (29), we obtain

$$\left\|\big(\mathbf{I} - (\mathbf{W}_{:,\mathbb{S}}^{(k-1)})^T\mathbf{D}_{:,\mathbb{S}}\big)(\mathbf{x}_{\mathbb{S}}^{(k-1)} - \mathbf{x}_{\mathbb{S}}^*) - \theta^{(k-1)}\text{sign}(\mathbf{x}_{\mathbb{S}}^{(k)})\right\|_2 = \|\mathbf{x}_{\mathbb{S}}^{(k)} - \mathbf{x}_{\mathbb{S}}^*\|_2 = \|\mathbf{x}^{(k)} - \mathbf{x}^*\|_2,$$

where the last equality is due to Definition 3. Thus, for all $\mathbf{x}^* \in \mathcal{X}^{(k)}(\epsilon)$, if case 1 holds, we have

$$\left\|\big(\mathbf{I} - (\mathbf{W}_{:,\mathbb{S}}^{(k-1)})^T\mathbf{D}_{:,\mathbb{S}}\big)(\mathbf{x}_{\mathbb{S}}^{(k-1)} - \mathbf{x}_{\mathbb{S}}^*) - \theta^{(k-1)}\text{sign}(\mathbf{x}_{\mathbb{S}}^{(k)})\right\|_2 \leq \epsilon \|\mathbf{x}^*\|_2 (\bar{\sigma}_{\min}/3^s)^k. \tag{30}$$

Multiplying both sides of (30) by $(\mathbf{I} - (\mathbf{W}_{:,\mathbb{S}}^{(k-1)})^T\mathbf{D}_{:,\mathbb{S}})^{-1}$, we have

$$\begin{aligned}
&\|\mathbf{x}_{\mathbb{S}}^{(k-1)} - \mathbf{x}_{\mathbb{S}}^* - \theta^{(k-1)}(\mathbf{I} - (\mathbf{W}_{:,\mathbb{S}}^{(k-1)})^T\mathbf{D}_{:,\mathbb{S}})^{-1}\text{sign}(\mathbf{x}_{\mathbb{S}}^{(k)})\|_2 \\
&\leq \|(\mathbf{I} - (\mathbf{W}_{:,\mathbb{S}}^{(k-1)})^T\mathbf{D}_{:,\mathbb{S}})^{-1}\|_2 \cdot \epsilon \|\mathbf{x}^*\|_2 (\bar{\sigma}_{\min}/3^s)^k \leq \epsilon \|\mathbf{x}^*\|_2 (\bar{\sigma}_{\min})^{k-1} 3^{-ks},
\end{aligned}$$

where the last inequality is due to (11). Let $\tilde{\mathbf{x}}^{(k-1)}$ denote the bias of $\mathbf{x}^{(k-1)}$:

$$\tilde{\mathbf{x}}^{(k-1)} \triangleq \theta^{(k-1)}(\mathbf{I} - (\mathbf{W}_{:,\mathbb{S}}^{(k-1)})^T\mathbf{D}_{:,\mathbb{S}})^{-1}\text{sign}(\mathbf{x}_{\mathbb{S}}^{(k)}),$$

then we get a condition that $\mathbf{x}^*$ satisfies if case 1 holds:

$$\mathcal{X}^{(k-1)}(\epsilon) = \left\{\mathbf{x}^* \Big| \left\|\mathbf{x}_{\mathbb{S}}^{(k-1)}(\mathbf{x}^*) - \mathbf{x}_{\mathbb{S}}^* - \tilde{\mathbf{x}}^{(k-1)}(\mathbf{x}^*)\right\|_2 \leq \epsilon \|\mathbf{x}^*\|_2 (\bar{\sigma}_{\min})^{k-1} 3^{-ks}\right\}.$$

If case 2 holds, $\mathbf{x}^*$ belongs to the following set:

$$\tilde{\mathcal{X}}^{(k)}(\epsilon) = \left\{\mathbf{x}^* \Big| |x_i^*| \leq \epsilon \|\mathbf{x}^*\|_2 \big(\bar{\sigma}_{\min}/3^s\big)^k, \text{ for some } i \in \mathbb{S}\right\}.$$

Then for any $\mathbf{x}^* \in \mathcal{X}^{(k)}(\epsilon)$, either $\mathbf{x}^* \in \mathcal{X}^{(k-1)}(\epsilon)$ or $\mathbf{x}^* \in \tilde{\mathcal{X}}^{(k)}(\epsilon)$ holds. In another word,

$$\mathcal{X}^{(k)}(\epsilon) \subset \tilde{\mathcal{X}}^{(k)}(\epsilon) \cup \mathcal{X}^{(k-1)}(\epsilon).$$

**Step 2: By imitating the construction of $\mathcal{X}^{(k)}(\epsilon)$, we construct**

$$\mathcal{X}^{(k-2)}(\epsilon), \mathcal{X}^{(k-3)}(\epsilon), \cdots.$$

Similar to Step 1, we divide $\mathcal{X}^{(k-1)}(\epsilon)$ into two sets: $\tilde{\mathcal{X}}^{(k-1)}(\epsilon)$ and $\mathcal{X}^{(k-2)}(\epsilon)$, then we divide $\mathcal{X}^{(k-2)}(\epsilon)$ into $\tilde{\mathcal{X}}^{(k-2)}(\epsilon)$ and $\mathcal{X}^{(k-3)}(\epsilon)$. Repeating the process, until dividing $\mathcal{X}^{(1)}(\epsilon)$ into $\tilde{\mathcal{X}}^{(1)}(\epsilon)$ and $\mathcal{X}^{(0)}(\epsilon)$.

By induction, we have

$$\mathcal{X}^{(k)}(\epsilon) \subset \tilde{\mathcal{X}}^{(k)}(\epsilon) \cup \tilde{\mathcal{X}}^{(k-1)}(\epsilon) \cup \tilde{\mathcal{X}}^{(k-2)}(\epsilon) \cup \cdots \cup \tilde{\mathcal{X}}^{(1)}(\epsilon) \cup \mathcal{X}^{(0)}(\epsilon), \quad (31)$$

where the sets are defined as follows for all $j = 0, 1, 2, \cdots, k$:

$$\tilde{\mathcal{X}}^{(k-j)}(\epsilon) = \left\{ \mathbf{x}^* \Big| |x_i^* + \tilde{x}_i^{(k-j)}(\mathbf{x}^*)| < \epsilon \|\mathbf{x}^*\|_2 (\bar{\sigma}_{\min})^{k-j} 3^{-ks}, \text{ for some } i \in \mathbb{S}. \right\}, \quad (32)$$

$$\mathcal{X}^{(k-j)}(\epsilon) = \left\{ \mathbf{x}^* \Big| \|\mathbf{x}_{\mathbb{S}}^{(k-j)}(\mathbf{x}^*) - \mathbf{x}_{\mathbb{S}}^* - \tilde{\mathbf{x}}^{(k-j)}(\mathbf{x}^*)\|_2 \le \epsilon \|\mathbf{x}^*\|_2 (\bar{\sigma}_{\min})^{k-j} 3^{-ks} \right\} \quad (33)$$

and the bias is defined as following for all $j = 0, 1, 2, \cdots, k$:

$$\tilde{\mathbf{x}}^{(k-j)}(\mathbf{x}^*) = \sum_{t=1}^{j} \left( \mathbf{I} - \left( \mathbf{W}_{:,\mathbb{S}}^{(k-j+t-1)} \right)^T \mathbf{D}_{:,\mathbb{S}} \right)^{-t} \theta^{(k-j+t-1)} \mathrm{sign}\big( \mathbf{x}_{\mathbb{S}}^{(k-j+t)}(\mathbf{x}^*) \big). \quad (34)$$

**Step 3: Estimating the probabilities of all the sets in (31).**

By (31), we have

$$P\Big( \mathbf{x}^* \in \mathcal{X}^{(k)}(\epsilon) \Big| \mathrm{support}(\mathbf{x}^*) = \mathbb{S} \Big)$$

$$\le \sum_{j=1}^{k-1} P\Big( \mathbf{x}^* \in \tilde{\mathcal{X}}^{(k-j)}(\epsilon) \Big| \mathrm{support}(\mathbf{x}^*) = \mathbb{S} \Big) + P\Big( \mathbf{x}^* \in \mathcal{X}^{(0)}(\epsilon) \Big| \mathrm{support}(\mathbf{x}^*) = \mathbb{S} \Big).$$

Now we have to prove that each of the above terms is small, then $P(\mathbf{x}^* \in \mathcal{X}^{(k)}(\epsilon) | \mathrm{support}(\mathbf{x}^*) = \mathbb{S})$ is small and (28) will be proved.

Define a set of $n$-dimensional sign numbers

$$\mathrm{Si}(n) = \Big\{ (s_1, s_2, \cdots, s_n) \Big| s_i \in \{0, -1, 1\}, \forall i = 1, \cdots, n \Big\}.$$

Since $\mathrm{sign}\big( \mathbf{x}_{\mathbb{S}}^{(k-j+t)} \big) \in \mathrm{Si}(|\mathbb{S}|)$ for all $t = 1, 2, \cdots, j$, $\{ \mathrm{sign}(\mathbf{x}_{\mathbb{S}}^{(k-j+t)}) \}_{t=1}^{j}$ has finitely possible values. Let $\mathrm{sign}(\mathbf{x}_{\mathbb{S}}^{(k-j+t)}) = \mathbf{s}^{(t)}$ for $t = 1, 2, \cdots, j$. Then $\tilde{x}_i^{(k-j)}(\mathbf{x}^*)$ is independent of $\mathbf{x}^*$ and can be written as $\tilde{x}_i^{(k-j)}(\mathbf{s}^{(1)}, \mathbf{s}^{(2)}, \cdots, \mathbf{s}^{(j)})$. Thus, we have

$$P(\mathbf{x}^* \in \tilde{\mathcal{X}}^{(k-j)}(\epsilon) | \mathrm{support}(\mathbf{x}^*) = \mathbb{S})$$

$$= \sum_{i \in \mathbb{S}} \sum_{\mathbf{s}^{(1)} \in \mathrm{Si}(|\mathbb{S}|)} \sum_{\mathbf{s}^{(2)} \in \mathrm{Si}(|\mathbb{S}|)} \cdots \sum_{\mathbf{s}^{(j)} \in \mathrm{Si}(|\mathbb{S}|)}$$

$$P\Big( |x_i^* + \tilde{x}_i^{(k-j)}(\mathbf{x}^*)| < \epsilon \|\mathbf{x}^*\|_2 (\bar{\sigma}_{\min})^{k-j} 3^{-ks}, \mathrm{sign}(\mathbf{x}_{\mathbb{S}}^{(k)}) = \mathbf{s}^{(1)}, \cdots, \mathrm{sign}(\mathbf{x}_{\mathbb{S}}^{(k-j+1)}) = \mathbf{s}^{(j)} \Big| \mathrm{support}(\mathbf{x}^*) = \mathbb{S} \Big)$$

$$\le \sum_{i \in \mathbb{S}} \sum_{\mathbf{s}^{(1)} \in \mathrm{Si}(|\mathbb{S}|)} \sum_{\mathbf{s}^{(2)} \in \mathrm{Si}(|\mathbb{S}|)} \cdots \sum_{\mathbf{s}^{(j)} \in \mathrm{Si}(|\mathbb{S}|)}$$

$$P\Big( |x_i^* + \tilde{x}_i^{(k-j)}(\mathbf{s}^{(1)}, \mathbf{s}^{(2)}, \cdots, \mathbf{s}^{(j)})| < \epsilon \sqrt{|\mathbb{S}|} B (\bar{\sigma}_{\min})^{k-j} 3^{-ks} \Big| \mathrm{support}(\mathbf{x}^*) = \mathbb{S} \Big)$$

$$\le \sum_{i \in \mathbb{S}} \sum_{\mathbf{s}^{(1)} \in \mathrm{Si}(|\mathbb{S}|)} \sum_{\mathbf{s}^{(2)} \in \mathrm{Si}(|\mathbb{S}|)} \cdots \sum_{\mathbf{s}^{(j)} \in \mathrm{Si}(|\mathbb{S}|)} \frac{\epsilon \sqrt{|\mathbb{S}|} B (\bar{\sigma}_{\min})^{k-j} 3^{-ks}}{B}$$

$$= |\mathbb{S}| 3^{j|\mathbb{S}|} (\epsilon \sqrt{|\mathbb{S}|} \big( (\bar{\sigma}_{\min})^{k-j} 3^{-ks} \big) \le \epsilon |\mathbb{S}|^{3/2} (\bar{\sigma}_{\min})^{k-j} 3^{(j-k)|\mathbb{S}|}$$

where the second inequality comes from the uniform distribution of $\mathbf{x}_{\mathbb{S}}^*$ (Assumption 2), the last inequality comes from $|\mathbb{S}| \le s$.

The last term, due to the uniform distribution of $\mathbf{x}_{\mathbb{S}}^*$ and $\mathbf{x}^{(0)} = \mathbf{0}$, can be bounded by

$$P(\mathbf{x}^* \in \mathcal{X}^{(0)}(\epsilon)|\text{support}(\mathbf{x}^*) = \mathbb{S})$$

$$=P\Big(\|\mathbf{x}^* + \tilde{\mathbf{x}}^{(0)}(\mathbf{x}^*)\|_2 \leq \epsilon\|\mathbf{x}^*\|_2 3^{-ks}\Big|\text{support}(\mathbf{x}^*) = \mathbb{S}\Big)$$

$$= \sum_{\mathbf{s}^{(1)} \in \text{Si}(|\mathbb{S}|)} \sum_{\mathbf{s}^{(2)} \in \text{Si}(|\mathbb{S}|)} \cdots \sum_{\mathbf{s}^{(k)} \in \text{Si}(|\mathbb{S}|)}$$

$$P\Big(\|\mathbf{x}^* + \tilde{\mathbf{x}}^{(0)}(\mathbf{x}^*)\|_2 \leq \epsilon\|\mathbf{x}^*\|_2 3^{-ks}, \text{sign}(\mathbf{x}_{\mathbb{S}}^{(1)}) = \mathbf{s}^{(1)}, \cdots, \text{sign}(\mathbf{x}_{\mathbb{S}}^{(k)}) = \mathbf{s}^{(k)}\Big|\text{support}(\mathbf{x}^*) = \mathbb{S}\Big)$$

$$\leq 3^{k|\mathbb{S}|}\Big((\epsilon 3^{-ks})^{|\mathbb{S}|}\Big) \leq \epsilon^{|\mathbb{S}|}.$$

Then we obtain

$$P(\mathbf{x}^* \in \mathcal{X}^{(k)}(\epsilon)|\text{support}(\mathbf{x}^*) = \mathbb{S})$$

$$\leq \sum_{j=0}^{k-1} \epsilon|\mathbb{S}|^{3/2}(\bar{\sigma}_{\min})^{k-j}3^{(j-k)|\mathbb{S}|} + \epsilon^{|\mathbb{S}|} = \sum_{j=1}^{k} \epsilon|\mathbb{S}|^{3/2}(\bar{\sigma}_{\min})^j 3^{-j|\mathbb{S}|} + \epsilon^{|\mathbb{S}|}$$

$$=\epsilon|\mathbb{S}|^{3/2}\frac{\bar{\sigma}_{\min}3^{-|\mathbb{S}|}}{1 - \bar{\sigma}_{\min}3^{-|\mathbb{S}|}}\Big(1 - (\bar{\sigma}_{\min}3^{-|\mathbb{S}|})^k\Big) + \epsilon^{|\mathbb{S}|} \leq \epsilon|\mathbb{S}|^{3/2} + \epsilon^{|\mathbb{S}|}.$$

Then (28) is proved. $\qquad\square$

## C  PROOF OF THEOREM 3

There are two conclusions in Theorem 3. We prove the two conclusions in the following two subsections respectively.

### C.1  PROOF OF CONCLUSION 1.

Before proving Conclusion 1, we analyze the operator $\mathbf{D}_{\text{cir}}^N$ in detail.

The circular convolution (23) is equivalent with:

$$\mathbf{b}(i,j) = \sum_{k=0}^{N-1}\sum_{l=0}^{N-1}\sum_{m=1}^{M}\mathbf{D}_{\text{cir}}^N(i,j;k,l,m)\mathbf{x}_m(k,l), \quad 0 \leq i,j \leq N-1,$$

where the circulant matrix is element-wise defined as:

$$\mathbf{D}_{\text{cir}}^N(i,j;k,l,m) = \begin{cases}\mathbf{d}_m\big((k-i)_{\text{mod}N}, (l-j)_{\text{mod}N}\big), & 0 \leq (k-i)_{\text{mod}N}, (l-j)_{\text{mod}N} \leq D-1 \\ 0, & \text{others}\end{cases}$$

$$(35)$$

Similarly, the corresponding circulant matrix $\mathbf{W}_{\text{cir}}^N(i,j;k,l,m)$ of dictionary $\mathbf{w}$ is:

$$\mathbf{W}_{\text{cir}}^N(i,j;k,l,m) = \begin{cases}\mathbf{w}_m\big((k-i)_{\text{mod}N}, (l-j)_{\text{mod}N}\big), & 0 \leq (k-i)_{\text{mod}N}, (l-j)_{\text{mod}N} \leq D-1 \\ 0, & \text{others}\end{cases}$$

$$(36)$$

As we defined in Section 3, $\mathbf{b}$ is a vector. With $\mathbf{x} = [\mathbf{x}_1, \cdots, \mathbf{x}_M]^T$, $\mathbf{x}$ is a vector. Then the operator $\mathbf{D}_{\text{cir}}^N$ is a matrix, where $(i,j)$ is its row index and $(k,l,m)$ is its column index.

Define a function measuring the difference between $i$ and $k$:

$$I(i,k) \triangleq (k-i)_{\text{mod}N}, \quad 0 \leq i,k \leq N-1.$$

The coherence between $\mathbf{D}_{\text{cir}}^N(i,j;k,l,m)$ and $\mathbf{W}_{\text{cir}}^N(i,j;k,l,m)$: $\mathbf{B}_{\text{coh}} = (\mathbf{D}_{\text{cir}}^N)^T\mathbf{W}_{\text{cir}}^N$ is element-wise defined by:

$$\mathbf{B}_{\text{coh}}(k_1,l_1,m_1;k_2,l_2,m_2) = \sum_{i=0}^{N-1}\sum_{j=0}^{N-1}\mathbf{D}_{\text{cir}}^N(i,j;k_1,l_1,m_1)\mathbf{W}_{\text{cir}}^N(i,j;k_2,l_2,m_2)$$

$$= \sum_{i\in\mathcal{I}(k_1,k_2)}\sum_{j\in\mathcal{J}(l_1,l_2)}\mathbf{d}_{m_1}\big(I(i,k_1),I(j,l_1)\big)\mathbf{w}_{m_2}\big(I(i,k_2),I(j,l_2)\big).$$

where

$$\mathcal{I}(k_1, k_2) = \{i | 0 \le i \le N - 1, \ 0 \le I(i, k_1) \le D - 1, \ 0 \le I(i, k_2) \le D - 1\},$$
$$\mathcal{J}(l_1, l_2) = \{j | 0 \le j \le N - 1, \ 0 \le I(j, l_1) \le D - 1, \ 0 \le I(j, l_2) \le D - 1\}.$$

**Lemma 2.** *Given $N \ge 2D - 1$, it holds that:*

*(a) $\mathcal{I}(k_1, k_2) \ne \varnothing$ if and only if "$0 \le (k_1 - k_2)_{\mathrm{mod}N} \le D - 1$" or "$0 < (k_2 - k_1)_{\mathrm{mod}N} \le D - 1$" holds.*

*(b) $\mathcal{J}(l_1, l_2) \ne \varnothing$ if and only if "$0 \le (l_1 - l_2)_{\mathrm{mod}N} \le D - 1$" or "$0 < (l_2 - l_1)_{\mathrm{mod}N} \le D - 1$" holds.*

*Proof.* Now we prove Conclusion (a). Firstly, we prove "if." If $0 \le (k_1 - k_2)_{\mathrm{mod}N} \le D - 1$ and $N \ge 2D - 1$, we have

$$\mathcal{I}(k_1, k_2) = \big\{(k_1 - \delta)_{\mathrm{mod}N} \big| \delta \in \mathbb{Z}, \ (k_1 - k_2)_{\mathrm{mod}N} \le \delta \le D - 1\big\} \ne \varnothing. \tag{37}$$

If $0 < (k_2 - k_1)_{\mathrm{mod}N} \le D - 1$ and $N \ge 2D - 1$, we have

$$\mathcal{I}(k_1, k_2) = \big\{(k_2 - \delta)_{\mathrm{mod}N} \big| \delta \in \mathbb{Z}, \ (k_2 - k_1)_{\mathrm{mod}N} \le \delta \le D - 1\big\} \ne \varnothing. \tag{38}$$

Secondly, we prove "only if." If $\mathcal{I}(k_1, k_2) \ne \varnothing$, we can select an $i \in \mathcal{I}(k_1, k_2)$. Let $r_1 = (k_1 - i)_{\mathrm{mod}N}$ and $r_2 = (k_2 - i)_{\mathrm{mod}N}$. By the definition of $\mathcal{I}(k_1, k_2)$, we have $0 \le r_1, r_2 \le D - 1$. Two cases should be considered here. Case 1: $r_1 \ge r_2$. Since $0 \le r_1 - r_2 \le D - 1 \le N - 1$, it holds that $r_1 - r_2 = (r_1 - r_2)_{\mathrm{mod}N}$. Thus,

$$\begin{aligned}
r_1 - r_2 = (r_1 - r_2)_{\mathrm{mod}N} &= \big((k_1 - i)_{\mathrm{mod}N} - (k_2 - i)_{\mathrm{mod}N}\big)_{\mathrm{mod}N} \\
&= \big((k_1 - i) - (k_2 - i)\big)_{\mathrm{mod}N} \\
&= (k_1 - k_2)_{\mathrm{mod}N}.
\end{aligned}$$

The equality "$0 \le r_1 - r_2 \le D - 1$" leads to the conclusion "$0 \le (k_1 - k_2)_{\mathrm{mod}N} \le D - 1$". In case 2 where $r_1 < r_2$, we can obtain $0 < (k_2 - k_1)_{\mathrm{mod}N} \le D - 1$ with the similar arguments.

Conclusion (b) can be proved by the same argument with the proof of (a). Lemma 2 is proved. $\square$

Now we fix $k_1, l_1$ and consider what values of $k_2, l_2$ give $\mathcal{I}(k_1, k_2) \ne \varnothing$ and $\mathcal{J}(l_1, l_2) \ne \varnothing$. Define four index sets given $0 \le k_1, l_1 \le N - 1$:

$$\begin{aligned}
\mathcal{K}(k_1) &= \{k | 0 \le (k_1 - k)_{\mathrm{mod}N} \le D - 1\} \\
\bar{\mathcal{K}}(k_1) &= \{k | 0 < (k - k_1)_{\mathrm{mod}N} \le D - 1\} \\
\mathcal{L}(l_1) &= \{l | 0 \le (l_1 - l)_{\mathrm{mod}N} \le D - 1\} \\
\bar{\mathcal{L}}(l_1) &= \{l | 0 < (l - l_1)_{\mathrm{mod}N} \le D - 1\}
\end{aligned}$$

**Lemma 3.** *If $N \ge 2D - 1$, we have:*

*(a) The cardinality of $\mathcal{K}(k_1), \bar{\mathcal{K}}(k_1)$: $|\mathcal{K}(k_1)| = D, |\bar{\mathcal{K}}(k_1)| = D - 1$.*

*(b) $\mathcal{K}(k_1) \cap \bar{\mathcal{K}}(k_1) = \varnothing$.*

*(c) The cardinality of $\mathcal{L}(l_1), \bar{\mathcal{L}}(l_1)$: $|\mathcal{L}(l_1)| = D, |\bar{\mathcal{L}}(l_1)| = D - 1$.*

*(d) $\mathcal{L}(l_1) \cap \bar{\mathcal{L}}(l_1) = \varnothing$.*

*Proof.* Now we prove Conclusion (a). The set $\mathcal{K}(k_1)$ can be equivalently written as

$$\mathcal{K}(k_1) = \{(k_1 - r_k)_{\mathrm{mod}N} | r_k = 0, 1, \cdots, D - 1\} \tag{39}$$

Let $k(r_k) = (k_1 - r_k)_{\mathrm{mod}N}$. We want to show that $k(r_k^1) \ne k(r_k^2)$ as long as $r_k^1 \ne r_k^2$. Without loss of generality, we assume $0 \le r_k^1 < r_k^2 \le D - 1$. By the definition of modulo operation, There exist two integers $q, q'$ such that

$$k(r_k^1) = qN + k_1 - r_k^1, \quad k(r_k^2) = q'N + k_1 - r_k^2.$$

Suppose $k(r_k^1) = k(r_k^2)$. Taking the difference between the above two equations, we obtain $r_k^2 - r_k^1 = (q' - q)N$, i.e, $N$ divides $r_k^2 - r_k^1$. However, $0 \le r_k^1 < r_k^2 \le D - 1$ implies $1 \le r_k^2 - r_k^1 \le D - 1 \le N - 1$, which contradicts with "$N$ dividing $r_k^2 - r_k^1$." Thus, it holds that $k(r_k^1) \ne k(r_k^2)$. Then we have $|\mathcal{K}(k_1)| = D$.

In the same way, we have

$$\bar{\mathcal{K}}(k_1) = \{(k_1 + r_k)_{\mathrm{mod}N} | r_k = 1, 2, \cdots, D - 1\} \tag{40}$$

and $|\bar{\mathcal{K}}(k_1)| = D - 1$. Conclusion (a) is proved.

Now we prove Conclusion (b). Suppose $\mathcal{K}(k_1) \cap \bar{\mathcal{K}}(k_1) \ne \varnothing$. Pick a $k_2 \in \mathcal{K}(k_1) \cap \bar{\mathcal{K}}(k_1)$. Let $r_3 = (k_1 - k_2)_{\mathrm{mod}N}$ and $r_4 = (k_2 - k_1)_{\mathrm{mod}N}$. Then we have $0 \le r_3 \le D - 1$ and $0 < r_4 \le D - 1$. By the definition of modulo operation, There exist two integers $q, q'$ such that

$$k_1 - k_2 = qN + r_3, \quad k_2 - k_1 = q'N + r_4$$

which imply

$$r_3 + r_4 + (q + q')N = 0.$$

However, $0 < r_3 + r_4 \le 2D - 2$ contradicts with "$q \in \mathbb{Z}, q' \in \mathbb{Z}, N \in \mathbb{Z}, N \ge 2D - 1$." Conclusion (b) is proved.

Conclusions (c) and (d) are actually the same with Conclusions (a) and (b) respectively. Thus, it holds that

$$\mathcal{L}(l_1) = \{(l_1 - r_l)_{\mathrm{mod}N} | r_l = 0, 1, \cdots, D - 1\} \tag{41}$$
$$\bar{\mathcal{L}}(l_1) = \{(l_1 + r_l)_{\mathrm{mod}N} | r_l = 1, 2, \cdots, D - 1\} \tag{42}$$

and $|\mathcal{L}(l_1)| = D, |\bar{\mathcal{L}}(l_1)| = D - 1$. Lemma 3 is proved. $\qquad\square$

With the preparations, we can prove Conclusion 1 of Theorem 3 now.

*Proof of Theorem 3, Conclusion 1.* Firstly we fix $k_1 \in \{0, 1, \cdots, N - 1\}$ and consider $k_2 \in \mathcal{K}(k_1)$. Let $r_k = (k_1 - k_2)_{\mathrm{mod}N}$. Then equation (37) implies that, for any $i \in \mathcal{I}(k_1, k_2)$, there exists a $\delta$ $(r_k \le \delta \le D - 1)$ such that

$$
\begin{aligned}
I(i, k_1) &= \big(k_1 - (k_1 - \delta)_{\mathrm{mod}N}\big)_{\mathrm{mod}N} = (\delta)_{\mathrm{mod}N} = \delta, \\
I(i, k_2) &= \big(k_2 - (k_1 - \delta)_{\mathrm{mod}N}\big)_{\mathrm{mod}N} = (\delta - r_k)_{\mathrm{mod}N} = \delta - r_k.
\end{aligned}
\tag{43}
$$

Now we consider another case for $k_2$: $k_2 \in \bar{\mathcal{K}}(k_1), r_k = (k_2 - k_1)_{\mathrm{mod}N}$. Equation (38) implies that, for any $i \in \mathcal{I}(k_1, k_2)$, there exists a $\delta$ $(r_k \le \delta \le D - 1)$ such that

$$
\begin{aligned}
I(i, k_1) &= \big(k_1 - (k_2 - \delta)_{\mathrm{mod}N}\big)_{\mathrm{mod}N} = (\delta - r_k)_{\mathrm{mod}N} = \delta - r_k, \\
I(i, k_2) &= \big(k_2 - (k_2 - \delta)_{\mathrm{mod}N}\big)_{\mathrm{mod}N} = (\delta)_{\mathrm{mod}N} = \delta.
\end{aligned}
\tag{44}
$$

Similarly, for any $l_1 \in \{0, 1, \cdots, N - 1\}$ and $l_2 \in \mathcal{L}(l_1)$, we denote $r_l = (l_1 - l_2)_{\mathrm{mod}N}$. For any $j \in \mathcal{J}(l_1, l_2)$, there exists a $\delta$ $(r_l \le \delta \le D - 1)$ such that

$$
\begin{aligned}
I(j, l_1) &= \big(l_1 - (l_1 - \delta)_{\mathrm{mod}N}\big)_{\mathrm{mod}N} = (\delta)_{\mathrm{mod}N} = \delta, \\
I(j, l_2) &= \big(l_2 - (l_1 - \delta)_{\mathrm{mod}N}\big)_{\mathrm{mod}N} = (\delta - r_l)_{\mathrm{mod}N} = \delta - r_l.
\end{aligned}
\tag{45}
$$

Another case for $l_2$: $l_2 \in \bar{\mathcal{L}}(l_1), r_l = (l_2 - l_1)_{\mathrm{mod}N}$. For any $j \in \mathcal{J}(l_1, l_2)$, there exists a $\delta$ $(r_l \le \delta \le D - 1)$ such that

$$
\begin{aligned}
I(j, l_1) &= \big(l_1 - (l_2 - \delta)_{\mathrm{mod}N}\big)_{\mathrm{mod}N} = (\delta - r_l)_{\mathrm{mod}N} = \delta - r_l, \\
I(j, l_2) &= \big(l_2 - (l_2 - \delta)_{\mathrm{mod}N}\big)_{\mathrm{mod}N} = (\delta)_{\mathrm{mod}N} = \delta.
\end{aligned}
\tag{46}
$$

Now let us consider the following function. By results in Lemmas 2 and 3, we have

$$
\begin{aligned}
f(k_1, l_1, m_1, m_2) &= \sum_{k_2=0}^{N-1} \sum_{l_2=0}^{N-1} \Big(\mathbf{B}_{\mathrm{coh}}(k_1, l_1, m_1; k_2, l_2, m_2)\Big)^2 \\
&= f_1 + f_2 + f_3 + f_4,
\end{aligned}
$$

where

$$f_1 = \sum_{k_2 \in \mathcal{K}(k_1)} \sum_{l_2 \in \mathcal{L}(l_1)} \Big( \mathbf{B}_{\mathrm{coh}}(k_1, l_1, m_1; k_2, l_2, m_2) \Big)^2$$

$$f_2 = \sum_{k_2 \in \bar{\mathcal{K}}(k_1)} \sum_{l_2 \in \mathcal{L}(l_1)} \Big( \mathbf{B}_{\mathrm{coh}}(k_1, l_1, m_1; k_2, l_2, m_2) \Big)^2$$

$$f_3 = \sum_{k_2 \in \mathcal{K}(k_1)} \sum_{l_2 \in \bar{\mathcal{L}}(l_1)} \Big( \mathbf{B}_{\mathrm{coh}}(k_1, l_1, m_1; k_2, l_2, m_2) \Big)^2$$

$$f_4 = \sum_{k_2 \in \bar{\mathcal{K}}(k_1)} \sum_{l_2 \in \bar{\mathcal{L}}(l_1)} \Big( \mathbf{B}_{\mathrm{coh}}(k_1, l_1, m_1; k_2, l_2, m_2) \Big)^2.$$

Combining equations (39), (41), (43) and (45), we obtain

$$f_1 = \sum_{r_k=0}^{D-1} \sum_{r_l=0}^{D-1} \sum_{\delta_k=r_k}^{D-1} \sum_{\delta_l=r_l}^{D-1} \Big( \mathbf{d}_{m_1}(\delta_k, \delta_l) \mathbf{w}_{m_2}(\delta_k - r_k, \delta_l - r_l) \Big)^2.$$

Combining (40), (41), (44) and (45), we obtain

$$f_2 = \sum_{r_k=1}^{D-1} \sum_{r_l=0}^{D-1} \sum_{\delta_k=r_k}^{D-1} \sum_{\delta_l=r_l}^{D-1} \Big( \mathbf{d}_{m_1}(\delta_k - r_k, \delta_l) \mathbf{w}_{m_2}(\delta_k, \delta_l - r_l) \Big)^2.$$

Combining (39), (42), (43) and (46), we obtain

$$f_3 = \sum_{r_k=0}^{D-1} \sum_{r_l=1}^{D-1} \sum_{\delta_k=r_k}^{D-1} \sum_{\delta_l=r_l}^{D-1} \Big( \mathbf{d}_{m_1}(\delta_k, \delta_l - r_l) \mathbf{w}_{m_2}(\delta_k - r_k, \delta_l) \Big)^2.$$

Combining (40), (42), (44) and (46), we obtain

$$f_4 = \sum_{r_k=1}^{D-1} \sum_{r_l=1}^{D-1} \sum_{\delta_k=r_k}^{D-1} \sum_{\delta_l=r_l}^{D-1} \Big( \mathbf{d}_{m_1}(\delta_k - r_k, \delta_l - r_l) \mathbf{w}_{m_2}(\delta_k, \delta_l) \Big)^2.$$

By the above explicit formulas of $f_i, 1 \le i \le 4$, we have $f_1, f_2, f_3, f_4$ are all independent of $k_1, l_1$ and $N$. They are only related with $m_1, m_2$ for fixed $\mathbf{d}$ and $\mathbf{m}$. Thus, we are able to denote $f(k_1, l_1, m_1, m_2)$ as $f(m_1, m_2)$ for simplicity. Consequently,

$$\begin{aligned} \frac{1}{N^2} \|(\mathbf{D}_{\mathrm{cir}}^N)^T \mathbf{W}_{\mathrm{cir}}^N\|_F^2 &= \frac{1}{N^2} \sum_{k_1=0}^{N-1} \sum_{l_1=0}^{N-1} \sum_{k_2=0}^{N-1} \sum_{l_2=0}^{N-1} \sum_{m_1=1}^{M} \sum_{m_2=1}^{M} \Big( \mathbf{B}_{\mathrm{coh}}(k_1, l_1, m_1; k_2, l_2, m_2) \Big)^2 \\ &= \frac{1}{N^2} \sum_{k_1=0}^{N-1} \sum_{l_1=0}^{N-1} \sum_{m_1=1}^{M} \sum_{m_2=1}^{M} f(k_1, l_1, m_1, m_2) \\ &= \frac{1}{N^2} \sum_{k_1=0}^{N-1} \sum_{l_1=0}^{N-1} \sum_{m_1=1}^{M} \sum_{m_2=1}^{M} f(m_1, m_2) \\ &= \frac{1}{N^2} \cdot N^2 \cdot \sum_{m_1=1}^{M} \sum_{m_2=1}^{M} f(m_1, m_2) = \sum_{m_1=1}^{M} \sum_{m_2=1}^{M} f(m_1, m_2) \end{aligned}$$

Thus, $\frac{1}{N^2} \|(\mathbf{D}_{\mathrm{cir}}^N)^T \mathbf{W}_{\mathrm{cir}}^N\|_F^2$ is dependent of $N$:

$$\frac{1}{N^2} \|(\mathbf{D}_{\mathrm{cir}}^N)^T \mathbf{W}_{\mathrm{cir}}^N\|_F^2 = \frac{1}{(2D-1)^2} \|(\mathbf{D}_{\mathrm{cir}}^{2D-1})^T \mathbf{W}_{\mathrm{cir}}^{2D-1}\|_F^2, \quad \forall N \ge 2D-1, \qquad (47)$$

which implies $\mathcal{W}_{\mathrm{cir}}^N = \mathcal{W}_{\mathrm{cir}}^{2D-1}, \forall N \ge 2D-1$. $\qquad\square$

## C.2 PROOF OF CONCLUSION 2.

Before proving Conclusion 2, let us analyze the relationship between $\mathbf{D}_{\text{conv}}^{N}$ and $\mathbf{D}_{\text{cir}}^{N+D-1}$.

Similar to $\mathbf{D}_{\text{cir}}$, we use $(i,j)$ as the row index and $(k,l,m)$ as the column index of $\mathbf{D}_{\text{conv}}$. For $0 \le i, j \le N - 1, 1 \le m \le M$,

$$\mathbf{D}_{\text{cir}}^{N+D-1}(i,j;k,l,m) = \mathbf{D}_{\text{conv}}^{N}(i,j;k,l,m) = \begin{cases} \mathbf{d}_m(k-i, l-j), & 0 \le k-i, l-j \le D-1 \\ 0, & k, l \text{ taken as others} \end{cases}$$

(48)

Matrix $\mathbf{D}_{\text{cir}}^{N+D-1}$ is of dimension $(N+D-1)^2 \times (N+D-1)^2 M$, where $0 \le i, j \le N+D-2$; matrix $\mathbf{D}_{\text{conv}}^{N}$ is of dimension $(N)^2 \times (N+D-1)^2 M$, where $0 \le i, j \le N-1$. Thus, $\mathbf{D}_{\text{conv}}^{N}$ is a block in $\mathbf{D}_{\text{cir}}^{N+D-1}$, i.e.,

$$\mathbf{D}_{\text{cir}}^{N+D-1} = \begin{bmatrix} \mathbf{D}_{\text{conv}}^{N} \\ \Delta_{\mathbf{D}}^{N} \end{bmatrix}.$$

The matrix $\Delta_{\mathbf{D}}^{N}$ is of dimension $((N+D-1)^2 - N^2) \times (N+D-1)^2 M$:

$$\Delta_{\mathbf{D}}^{N} = \left[ \mathbf{D}_{\text{cir}}^{N+D-1}(i,j;:,:,:) \right], \quad (i,j) \in \mathcal{I}_\Delta$$

where

$$\begin{aligned} \mathcal{I}_\Delta &= \mathcal{I}_1 \cup \mathcal{I}_2 \cup \mathcal{I}_3 \\ \mathcal{I}_1 &= \{(i,j) | N \le i \le N+D-2, \ 0 \le j \le N-1\} \\ \mathcal{I}_2 &= \{(i,j) | 0 \le i \le N-1, \ N \le j \le N+D-2\} \\ \mathcal{I}_3 &= \{(i,j) | N \le i \le N+D-2, \ N \le j \le N+D-2\}. \end{aligned}$$

Similarly,

$$\mathbf{W}_{\text{cir}}^{N+D-1} = \begin{bmatrix} \mathbf{W}_{\text{conv}}^{N} \\ \Delta_{\mathbf{W}}^{N} \end{bmatrix}, \quad \Delta_{\mathbf{W}}^{N} = \left[ \mathbf{W}_{\text{cir}}^{N+D-1}(i,j;:,:,:) \right], \quad (i,j) \in \mathcal{I}_\Delta.$$

Then,

$$(\mathbf{D}_{\text{cir}}^{N+D-1})^T \mathbf{W}_{\text{cir}}^{N+D-1} = (\mathbf{D}_{\text{conv}}^{N})^T \mathbf{W}_{\text{conv}}^{N} + (\Delta_{\mathbf{D}}^{N})^T \Delta_{\mathbf{W}}^{N}.$$

(49)

**Lemma 4.** *For any $(i,j) \in \mathcal{I}_\Delta$, one has*

$$\|\mathbf{D}_{\text{cir}}^{N+D-1}(i,j;:,:,:)\|_2^2 = \|\mathbf{d}\|_2^2,$$

(50)

$$\|\mathbf{W}_{\text{cir}}^{N+D-1}(i,j;:,:,:)\|_2^2 = \|\mathbf{w}\|_2^2.$$

(51)

*Proof.* Equation (35) implies that, for $(i,j) \in \mathcal{I}_1, 1 \le m \le M$,

$$\mathbf{D}_{\text{cir}}^{N+D-1}(i,j;k,l,m) = \begin{cases} \mathbf{d}_m(k-i, l-j), & i \le k \le N+D-2, j \le l \le j+D-1 \\ \mathbf{d}_m(k-i+N+D-1, l-j), & 0 \le k \le i-N, j \le l \le j+D-1 \\ 0, & k, l \text{ taken as others} \end{cases}$$

Thus, for any $(i,j) \in \mathcal{I}_1$,

$$\|\mathbf{D}_{\text{cir}}^{N+D-1}(i,j;:,:,:)\|_2^2 = \sum_{k=0}^{N+D-2} \sum_{l=0}^{N+D-2} \sum_{m=1}^{M} \left| \mathbf{D}_{\text{cir}}^{N+D-1}(i,j;k,l,m) \right|^2 = \|\mathbf{d}\|_2^2$$

Similarly,

$$\|\mathbf{D}_{\text{cir}}^{N+D-1}(i,j;:,:,:)\|_2^2 = \|\mathbf{d}\|_2^2, \quad (i,j) \in \mathcal{I}_2 \cup \mathcal{I}_3.$$

Equation (50) is proved. With the same argument, equation (51) is also proved. $\qquad\square$

**Lemma 5.** *If $N \ge 2D - 1$, we have*

$$\|(\Delta_{\mathbf{D}}^{N})^T \Delta_{\mathbf{W}}^{N}\|_F^2 \le \left(2N(D-1) + (D-1)^2\right)(2D-1)^2 \|\mathbf{d}\|_2^2 \|\mathbf{w}\|_2^2.$$

(52)

*Proof.* For simplicity, we denote two row vectors:

$$\mathbf{d}_{i,j} \triangleq \mathbf{D}_{\mathrm{cir}}^{N+D-1}(i,j;:,:,:) \in \mathbb{R}^{1\times(N+D-1)^2 M}$$
$$\mathbf{w}_{i,j} \triangleq \mathbf{W}_{\mathrm{cir}}^{N+D-1}(i,j;:,:,:) \in \mathbb{R}^{1\times(N+D-1)^2 M}$$

Then,

$$\|(\Delta_{\mathbf{D}}^N)^T \Delta_{\mathbf{W}}^N\|_F^2 = \left\| \sum_{(i,j)\in\mathcal{I}_\Delta} \mathbf{d}_{i,j}^T \mathbf{w}_{i,j} \right\|_F^2 = \sum_{(i_1,j_1)\in\mathcal{I}_\Delta} \sum_{(i_2,j_2)\in\mathcal{I}_\Delta} \left\langle \mathbf{d}_{i_1,j_1}^T \mathbf{w}_{i_1,j_1}, \mathbf{d}_{i_2,j_2}^T \mathbf{w}_{i_2,j_2} \right\rangle_F,$$

where

$$\left\langle \mathbf{d}_{i_1,j_1}^T \mathbf{w}_{i_1,j_1}, \mathbf{d}_{i_2,j_2}^T \mathbf{w}_{i_2,j_2} \right\rangle_F = \mathrm{trace}\left( \mathbf{w}_{i_1,j_1}^T \mathbf{d}_{i_1,j_1} \mathbf{d}_{i_2,j_2}^T \mathbf{w}_{i_2,j_2} \right) = (\mathbf{d}_{i_1,j_1} \mathbf{d}_{i_2,j_2}^T) \cdot (\mathbf{w}_{i_1,j_1} \mathbf{w}_{i_2,j_2}^T).$$

Since

$$\mathbf{d}_{i_1,j_1} \mathbf{d}_{i_2,j_2}^T = \sum_{k=0}^{N-1} \sum_{l=0}^{N-1} \sum_{m=1}^{M} = \mathbf{D}_{\mathrm{cir}}^{N+D-1}(i_1,j_1;k,l,m) \mathbf{D}_{\mathrm{cir}}^{N+D-1}(i_2,j_2;k,l,m),$$

with the same argument in Lemma 2, we have: $\mathbf{d}_{i_1,j_1} \mathbf{d}_{i_2,j_2}^T \neq 0$ implies

$$i_2 \in \mathcal{I}_\Delta' \triangleq \{i | 0 \leq (i_1-i)_{\mathrm{mod}(N+D-1)} \leq D-1 \text{ or } 0 \leq (i-i_1)_{\mathrm{mod}(N+D-1)} \leq D-1\}$$

$$j_2 \in \mathcal{J}_\Delta' \triangleq \{j | 0 \leq (j_1-j)_{\mathrm{mod}(N+D-1)} \leq D-1 \text{ or } 0 \leq (j-j_1)_{\mathrm{mod}(N+D-1)} \leq D-1\}$$

Then

$$\begin{aligned}
\|(\Delta_{\mathbf{D}}^N)^T \Delta_{\mathbf{W}}^N\|_F^2 &= \sum_{(i_1,j_1)\in\mathcal{I}_\Delta} \sum_{i_2\in\mathcal{I}_\Delta'} \sum_{j_2\in\mathcal{J}_\Delta'} (\mathbf{d}_{i_1,j_1} \mathbf{d}_{i_2,j_2}^T) \cdot (\mathbf{w}_{i_1,j_1} \mathbf{w}_{i_2,j_2}^T) \\
&\leq \sum_{(i_1,j_1)\in\mathcal{I}_\Delta} \sum_{i_2\in\mathcal{I}_\Delta'} \sum_{j_2\in\mathcal{J}_\Delta'} \|\mathbf{d}\|_2^2 \|\mathbf{w}\|_2^2 \\
&= |\mathcal{I}_\Delta| \cdot |\mathcal{I}_\Delta'| \cdot |\mathcal{J}_\Delta'| \cdot \|\mathbf{d}\|_2^2 \|\mathbf{w}\|_2^2 \\
&= \left(2N(D-1) + (D-1)^2\right)(2D-1)^2 \|\mathbf{d}\|_2^2 \|\mathbf{w}\|_2^2,
\end{aligned}$$

where the inequality in the second line follows from (50) and (51). Inequality (52) is proved. $\qquad\square$

With these preparations, we can prove Theorem 3, Conclusion 2 now.

*Proof of Theorem 3, Conclusion 2.* Define set

$$\mathcal{W}_{\mathrm{normal}} = \left\{ \mathbf{w} \in \mathbb{R}^{D^2 M} \middle| \mathbf{w}_m \cdot \mathbf{d}_m = 1, \ \forall m = 1, \cdots, M \right\}. \tag{53}$$

Since $\mathbf{d} \in \mathcal{W}_{\mathrm{normal}}$, the set is nonempty:

$$\mathcal{W}_{\mathrm{normal}} \neq \varnothing. \tag{54}$$

Define functions $F_{\mathrm{conv}}^N : \mathbb{R}^{D^2 M} \to \mathbb{R}$, $F_{\mathrm{cir}}^N : \mathbb{R}^{D^2 M} \to \mathbb{R}$.

$$F_{\mathrm{conv}}^N(\mathbf{w}) = \frac{1}{N+D-1} \left\| (\mathbf{D}_{\mathrm{conv}}^N(\mathbf{d}))^T \mathbf{W}_{\mathrm{conv}}^N(\mathbf{w}) \right\|_F + \iota_{\mathcal{W}_{\mathrm{normal}}}(\mathbf{w})$$

$$F_{\mathrm{cir}}^N(\mathbf{w}) = \frac{1}{N} \left\| (\mathbf{D}_{\mathrm{cir}}^N(\mathbf{d}))^T \mathbf{W}_{\mathrm{cir}}^N(\mathbf{w}) \right\|_F + \iota_{\mathcal{W}_{\mathrm{normal}}}(\mathbf{w})$$

By the definitions of $\mathcal{W}_{\mathrm{conv}}^N, \mathcal{W}_{\mathrm{cir}}^N$, we have

$$\mathcal{W}_{\mathrm{conv}}^N = \arg\min_{\mathbf{w}} F_{\mathrm{conv}}^N(\mathbf{w}), \quad \mathcal{W}_{\mathrm{cir}}^N = \arg\min_{\mathbf{w}} F_{\mathrm{cir}}^N(\mathbf{w})$$

**Step 1:** Proving $F_{\text{conv}}^N(\mathbf{w})$ uniformly converges to $F_{\text{cir}}^{2D-1}(\mathbf{w})$ on $X \cap \mathcal{W}_{\text{normal}}$ for any compact set $X \subset \mathbb{R}^{D^2 M}$.

We arbitrarily choose such a compact set $X$. Based on (47), (49) and (52), one has, for all $\mathbf{w} \in X \cap \mathcal{W}_{\text{normal}}$,

$$
\begin{aligned}
|F_{\text{conv}}^N(\mathbf{w}) - F_{\text{cir}}^{2D-1}(\mathbf{w})| &= |F_{\text{conv}}^N(\mathbf{w}) - F_{\text{cir}}^{N+D-1}(\mathbf{w})| \\
&= \frac{1}{N+D-1} \left| \left\| (\mathbf{D}_{\text{cir}}^{N+D-1})^T \mathbf{W}_{\text{cir}}^{N+D-1} \right\|_F - \left\| (\mathbf{D}_{\text{conv}}^N)^T \mathbf{W}_{\text{conv}}^N \right\|_F \right| \\
&\leq \frac{1}{N+D-1} \left\| (\Delta_{\mathbf{D}}^N)^T \Delta_{\mathbf{W}}^N \right\|_F \\
&\leq \frac{\sqrt{\left(2N(D-1) + (D-1)^2\right)(2D-1)}}{N+D-1} \|\mathbf{d}\|_2 \|\mathbf{w}\|_2 \\
&\leq \frac{(2D-1)\sqrt{2(D-1)}}{\sqrt{N+D-1}} \|\mathbf{d}\|_2 \|\mathbf{w}\|_2.
\end{aligned}
$$

Thus, there exists a constant $B > 0$, which is independent of $N$, such that

$$
|F_{\text{conv}}^N(\mathbf{w}) - F_{\text{cir}}^{2D-1}(\mathbf{w})| \leq \frac{B}{\sqrt{N}} \sup_{\mathbf{w} \in X \cap \mathcal{W}_{\text{normal}}} \|\mathbf{w}\|, \quad \forall \, \mathbf{w} \in X \cap \mathcal{W}_{\text{normal}}. \tag{55}
$$

**Step 2:** Proving $F_{\text{conv}}^N(\mathbf{w})$ epigraphically converges[8] to $F_{\text{cir}}^{2D-1}(\mathbf{w})$.

We want to show, at each point $\mathbf{w}$ it holds that

$$
\liminf_{N \to \infty} F_{\text{conv}}^N(\mathbf{w}^N) \geq F_{\text{cir}}^{2D-1}(\mathbf{w}) \quad \text{for every sequence } \mathbf{w}^N \to \mathbf{w} \tag{56}
$$

$$
\limsup_{N \to \infty} F_{\text{conv}}^N(\mathbf{w}^N) \leq F_{\text{cir}}^{2D-1}(\mathbf{w}) \quad \text{for some sequence } \mathbf{w}^N \to \mathbf{w} \tag{57}
$$

Firstly, we prove (56). We arbitrarily pick a sequence $\{\mathbf{w}^N\}_{N=0}^\infty$ such that $\mathbf{w}^N \to \mathbf{w}$.

If $\mathbf{w} \notin \mathcal{W}_{\text{normal}}$, $F_{\text{cir}}^{2D-1}(\mathbf{w}) = +\infty$. Since $\mathcal{W}_{\text{normal}}$ is a closed set, there exists a $N^+$ such that $\mathbf{w}^N \notin \mathcal{W}_{\text{normal}}$ for all $N \geq N^+$. Thus, one has $F_{\text{conv}}^N(\mathbf{w}^N) = +\infty$ for all $N \geq N^+$, i.e.,

$$
\liminf_{N \to \infty} F_{\text{conv}}^N(\mathbf{w}^N) = F_{\text{cir}}^{2D-1}(\mathbf{w}) = +\infty.
$$

If $\mathbf{w} \in \mathcal{W}_{\text{normal}}$, two cases should be considered. The first case is that any subsequences of $\{\mathbf{w}^N\}_{N=0}^\infty$ are not kept within $\mathcal{W}_{\text{normal}}$, i.e., there exists a $N^+$ such that $\mathbf{w}^N \notin \mathcal{W}_{\text{normal}}$ for all $N \geq N^+$. Then we have

$$
\liminf_{N \to \infty} F_{\text{conv}}^N(\mathbf{w}^N) = +\infty > F_{\text{cir}}^{2D-1}(\mathbf{w}).
$$

The second case is that there exists a subsequence $\{\mathbf{w}^{N_k}\}_{k=0}^\infty \subset \{\mathbf{w}^N\}_{N=0}^\infty$ such that

$$
\mathbf{w}^{N_k} \in \mathcal{W}_{\text{normal}}, \quad \forall k = 0, 1, 2, \cdots.
$$

Since $\mathbf{w}^N$ converges to $\mathbf{w}$, any subsequences should be Cauchy. Given any Cauchy sequence $\{\mathbf{w}^{N_k}\}_{k=0}^\infty$ in finite dimensional Euclidean space, there exists a compact set $X$ such that

$$
\mathbf{w}^{N_k} \in X, \quad \forall k = 0, 1, 2, \cdots
$$

Let $B' = \sup_{\mathbf{w} \in X \cap \mathcal{W}_{\text{normal}}} \|\mathbf{w}\|$. By (55), we obtain

$$
\begin{aligned}
|F_{\text{conv}}^{N_k}(\mathbf{w}^{N_k}) - F_{\text{cir}}^{2D-1}(\mathbf{w})| &\leq |F_{\text{conv}}^{N_k}(\mathbf{w}^{N_k}) - F_{\text{cir}}^{2D-1}(\mathbf{w}^{N_k})| + |F_{\text{cir}}^{2D-1}(\mathbf{w}^{N_k}) - F_{\text{cir}}^{2D-1}(\mathbf{w})| \\
&\leq \frac{BB'}{\sqrt{N_k}} + |F_{\text{cir}}^{2D-1}(\mathbf{w}^{N_k}) - F_{\text{cir}}^{2D-1}(\mathbf{w})|.
\end{aligned}
$$

---

[8]Epigraphic convergence is a standard tool to prove the convergence of a sequence of minimization problems. The definition of epigraphic convergence refers to Definition 7.1 and Proposition 7.2 in (Rockafellar & Wets, 2009).

For any $\epsilon > 0$, by the continuity of $F_{\text{cir}}^{2D-1}$, we are able to find a $K > 0$ such that $|F_{\text{cir}}^{2D-1}(\mathbf{w}^{N_k}) - F_{\text{cir}}^{2D-1}(\mathbf{w})| < \epsilon$ for all $k \geq K$. Pick a $K'$ such that $N_{K'} \geq (BB'/\epsilon)^2$. Then, for all $k \geq \max(K, K')$, we have $|F_{\text{conv}}^{N_k}(\mathbf{w}^{N_k}) - F_{\text{cir}}^{2D-1}(\mathbf{w})| < 2\epsilon$, i.e.,

$$\lim_{k\to\infty} F_{\text{conv}}^{N_k}(\mathbf{w}^{N_k}) = F_{\text{cir}}^{2D-1}(\mathbf{w}).$$

The above conclusion holds for all subsequences $\{\mathbf{w}^{N_k}\}_{k=0}^{\infty} \subset \mathcal{W}_{\text{normal}}$. $F_{\text{cir}}^{2D-1}(\mathbf{w})$ is an accumulation point of $\{F_{\text{conv}}^{N}(\mathbf{w}^{N})\}_{N=0}^{\infty}$. All the other accumulation points of $\{F_{\text{conv}}^{N}(\mathbf{w}^{N})\}_{N=0}^{\infty}$ must be $+\infty$ because $F_{\text{conv}}^{N}(\mathbf{w}) = F_{\text{cir}}^{2D-1}(\mathbf{w}) = +\infty$ for all $\mathbf{w} \notin \mathcal{W}_{\text{normal}}$. Thus,

$$\liminf_{N\to\infty} F_{\text{conv}}^{N}(\mathbf{w}^{N}) = F_{\text{cir}}^{2D-1}(\mathbf{w}) < +\infty.$$

Secondly, we prove (57). We set $\mathbf{w}^{N} = \mathbf{w}$ for all $N = 0, 1, 2, \cdots$. Then (57) is a direct result of (55).

**Step 3: proving (25).**  Define

$$G(\mathbf{w}) = \left\| (\mathbf{D}_{\text{cir}}^{2D-1})^{T} \mathbf{W}_{\text{cir}}^{2D-1} \right\|_{F}^{2}.$$

We want to show that $G(\mathbf{w})$ is strongly convex.

Let $\tilde{\mathbf{w}}_i \in \mathbb{R}^{(2D-1)^2}$ be the $i^{\text{th}}$ column of $\mathbf{W}_{\text{cir}}^{2D-1}$, i.e.,

$$\mathbf{W}_{\text{cir}}^{2D-1} = \left[ \tilde{\mathbf{w}}_1, \tilde{\mathbf{w}}_2, \cdots, \tilde{\mathbf{w}}_{(2D-1)^2 M} \right]$$

Then

$$G(\mathbf{w}) = \sum_{i=1}^{(2D-1)^2 M} (\tilde{\mathbf{w}}_i)^{T} \left( \mathbf{D}_{\text{cir}}^{2D-1} (\mathbf{D}_{\text{cir}}^{2D-1})^{T} \right) \tilde{\mathbf{w}}_i.$$

Let $\tilde{\mathbf{w}} \in \mathbb{R}^{(2D-1)^4 M}$ vectorize $\mathbf{W}_{\text{cir}}^{2D-1}$, i.e.,

$$\tilde{\mathbf{w}} = \left[ (\tilde{\mathbf{w}}_1)^{T}, (\tilde{\mathbf{w}}_2)^{T}, \cdots, (\tilde{\mathbf{w}}_{(2D-1)^2 M})^{T} \right]^{T}.$$

Then $G(\mathbf{w})$ can be written as a quadratic form of $\tilde{\mathbf{w}}$:

$$G(\mathbf{w}) = \tilde{\mathbf{w}}^{T} Q \tilde{\mathbf{w}},$$

where

$$Q = \underbrace{\left[ \begin{array}{ccc} \left( \mathbf{D}_{\text{cir}}^{2D-1} (\mathbf{D}_{\text{cir}}^{2D-1})^{T} \right) & & \\ & \cdots & \\ & & \left( \mathbf{D}_{\text{cir}}^{2D-1} (\mathbf{D}_{\text{cir}}^{2D-1})^{T} \right) \end{array} \right]}_{\text{totally } (2D-1)^2 M \text{ diagonal blocks}}.$$

As long as at least one of the matrices $\{\mathbf{D}_{\text{cir},0}^{2D-1}, \cdots, \mathbf{D}_{\text{cir},M-1}^{2D-1}\}$ is non-singular, $\mathbf{D}_{\text{cir}}^{2D-1}$ is full row rank, which implies that $\mathbf{D}_{\text{cir}}^{2D-1} (\mathbf{D}_{\text{cir}}^{2D-1})^{T}$ is non-singular. Then $Q$ is positive definite.

The transform between $\mathbf{w}$ and $\tilde{\mathbf{w}}$ is linear. We denote the transform as $T$, i.e.,

$$\tilde{\mathbf{w}} = T\mathbf{w}.$$

It's trivial that $\|\tilde{\mathbf{w}}\|_2^2 = 0$ implies $\|\mathbf{W}_{\text{cir}}^{2D-1}\|_F^2 = 0$. By the definition of $\mathbf{W}_{\text{cir}}^{2D-1}$, $\|\mathbf{W}_{\text{cir}}^{2D-1}\|_F^2 = 0$ implies $\|\mathbf{w}\|_2^2 = 0$. Thus, linear operator $T$ is full column rank. Thus, $T^{T} Q T$ is positive definite, and

$$G(\mathbf{w}) = \mathbf{w}^{T} (T^{T} Q T) \mathbf{w}$$

is strongly convex. Then $F_{\text{cir}}^{2D-1}(\mathbf{w}) = \sqrt{G(\mathbf{w})} + \iota_{\mathcal{W}_{\text{normal}}}(\mathbf{w})$ has only one minimizer, i.e., $\mathcal{W}_{\text{cir}}^{2D-1}$ involves only a unique element.

Now we check the conditions of Propositions 7.32(c) and 7.33 in (Rockafellar & Wets, 2009) to apply them.

1. $F_{\text{conv}}^N \xrightarrow{\text{e}} F_{\text{cir}}^{2D-1}$. This is proved in Step 2.

2. $F_{\text{cir}}^{2D-1}$ is level bounded. Since $G(\mathbf{w})$ is strongly convex, $F_{\text{cir}}^{2D-1}(\mathbf{w}) = \sqrt{G(\mathbf{w})} + \iota_{\mathcal{W}_{\text{normal}}}(\mathbf{w})$ must be level bounded.

3. $F_{\text{cir}}^{2D-1} \not\equiv +\infty$. Since $\mathcal{W}_{\text{normal}}$ is nonempty (54), $\operatorname{dom} F_{\text{cir}}^{2D-1} \neq \varnothing$, $F_{\text{cir}}^{2D-1}$ is not constantly $+\infty$.

4. All the level set of $F_{\text{conv}}^N$ are connected. This can be derived from the convexity of $F_{\text{conv}}^N$.

5. $F_{\text{cir}}^{2D-1}$ and $F_{\text{conv}}^N$ are all lower semi-continuous and proper. This condition follows from the fact that the functions $F_{\text{cir}}^{2D-1}$ and $F_{\text{conv}}^N$ are all continuous functions defined on a nonempty closed convex domain $\mathcal{W}_{\text{normal}}$.

Applying Proposition 7.32(c), we have $\{F_{\text{conv}}^N\}$ is eventually level bounded. If we arbitrarily pick a $\mathbf{w}^N \in \mathcal{W}_{\text{conv}}^N$ and let $\mathbf{w}_{\text{cir}}$ be the unique point in $\mathcal{W}_{\text{cir}}^{2D-1}$. Applying Proposition 7.33, we have $\mathbf{w}^N \to \mathbf{w}_{\text{cir}}$. By Definition 4.1 in (Rockafellar & Wets, 2009), we obtain the convergence of the sequence of sets $\{\mathcal{W}_{\text{conv}}^N\}$: $\lim_{N \to \infty} \mathcal{W}_{\text{conv}}^N = \mathcal{W}_{\text{cir}}^{2D-1}$. $\qquad\square$

## D    DISCUSSION OF DEFINITION 2 (11)

In this section, we want to numerically show that, given typical $\mathbf{D}$ and $s$, there is a $\bar{\sigma}_{\min} > 0$ such that a random generated matrix $\mathbf{W} \in \bar{\mathcal{W}}(\mathbf{D}, s, \bar{\sigma}_{\min})$. However, given $\mathbf{D}$ and $\mathbf{W}$, it's intractable to completely check (11):

$$\sigma_{\min}\left(\mathbf{I} - (\mathbf{W}_{:,\mathbb{S}})^T \mathbf{D}_{:,\mathbb{S}}\right) \geq \bar{\sigma}_{\min}, \forall \mathbb{S} \text{ with } 2 \leq |\mathbb{S}| \leq s.$$

The reason is that there are extremely large amount of possible $\mathbb{S}$ s. For example, we take $M = 250, N = 500, s = 50$. There are totally

$$\binom{500}{50} + \binom{500}{49} + \cdots + \binom{500}{2}$$

possible $\mathbb{S}$s satisfying $2 \leq |\mathbb{S}| \leq s$. It's impossible to check (11) on all possible $\mathbb{S}$s.

Instead of checking all possible $\mathbb{S}$s, we sample 5000 $\mathbb{S}$s from the whole set:

$$\mathcal{S}' \subset \mathcal{S} = \{\mathbb{S} : \mathbb{S} \subset \{1, 2, \cdots, 500\} | 2 \leq |S| \leq s\},$$

where $\mathcal{S}'$ is the set of all the samples. Then we estimate $\bar{\sigma}_{\min}$ with the following quantity:

$$\bar{\sigma}'(\mathbf{D}, \mathbf{W}) = \min_{\mathbb{S} \in \mathcal{S}'} \left\{ \sigma_{\min}\left(\mathbf{I} - (\mathbf{W}_{:,\mathbb{S}})^T \mathbf{D}_{:,\mathbb{S}}\right) \right\}$$

Furthermore, we use the same $\mathbf{D}$ as that in Section 5 and generate 1000 $\mathbf{W}$s with each entry i.i.d sampled from the normal distribution. Then we normalize each column of the generated $\mathbf{W}$s. This technique is commonly used in sparse coding. Finally, we report the distribution of $\bar{\sigma}'(\mathbf{D}, \mathbf{W})$ with the fixed $\mathbf{D}$ and the 1000 sampled $\mathbf{W}$s in Figure 5.

Figure 5 demonstrates that, with the fixed $\mathbf{D}$, most of the random generated $\mathbf{W}$s have a $\bar{\sigma}'(\mathbf{D}, \mathbf{W})$ within the interval $[0.25, 0.35]$. Thus, the numerical results support our claim: with high probability, a random generated $\mathbf{W}$ satisfies

$$\min_{\mathbb{S} \in \mathcal{S}} \left\{ \sigma_{\min}\left(\mathbf{I} - (\mathbf{W}_{:,\mathbb{S}})^T \mathbf{D}_{:,\mathbb{S}}\right) \right\} \geq \bar{\sigma}_{\min} > 0,$$

that is, $\mathbf{W} \in \bar{\mathcal{W}}(\mathbf{D}, s, \bar{\sigma}_{\min})$.

## E    EFFICIENT ALGORITHM TO CALCULATE ANALYTIC WEIGHTS

### E.1    AN EFFICIENT ALGORITHM TO SOLVE (16)

In this section, we introduce an algorithm to solve (16) (we copy (16) below to facilitate reading):

$$\min_{\mathbf{W} \in \mathbb{R}^{N \times M}} \left\| \mathbf{W}^T \mathbf{D} \right\|_F^2, \quad \text{s.t. } (\mathbf{W}_{:,m})^T \mathbf{D}_{:,m} = 1, \forall m = 1, 2, \cdots, M,$$

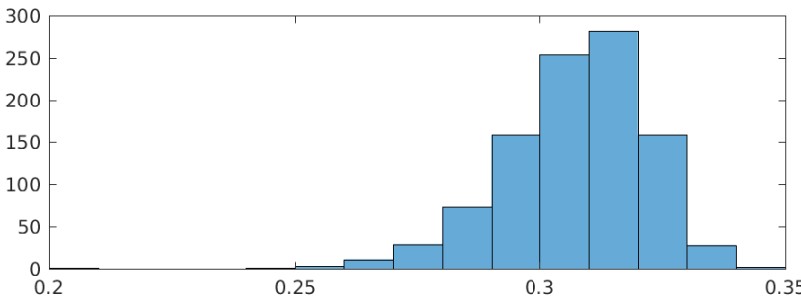

Figure 5: Discussion of Definition 2: distribution of $\bar{\sigma}'(\mathbf{D}, \mathbf{W})$ on random generated $\mathbf{W}$s.

By the definition of the Frobenius norm, it holds that

$$\|\mathbf{W}^T\mathbf{D}\|_F^2 = \|(\mathbf{W}^T\mathbf{D})^T\|_F^2 = \|\mathbf{D}^T\mathbf{W}\|_F^2. \tag{58}$$

Thus, the above problem is equivalent with

$$\min_{\mathbf{W} \in \mathbb{R}^{N \times M}} \left\|\mathbf{D}^T\mathbf{W}\right\|_F^2, \quad \text{s.t. } (\mathbf{D}_{:,m})^T\mathbf{W}_{:,m} = 1, \ \forall m = 1, 2, \cdots, M.$$

We apply projected gradient descent (PGD) to solve the above problem. The gradient of $\|\mathbf{D}^T\mathbf{W}\|_F^2$ is $\nabla\|\mathbf{D}^T\mathbf{W}\|_F^2 = \mathbf{D}\mathbf{D}^T\mathbf{W}$. Denote the set by

$$\mathcal{W} = \{\mathbf{W} \in \mathbb{R}^{N \times M} | (\mathbf{D}_{:,m})^T\mathbf{W}_{:,m} = 1, \ \forall m = 1, 2, \cdots, M.\}$$

Then the projection onto $\mathcal{W}$ can be calculated by

$$\text{Proj}_{\mathcal{W}}(\mathbf{W}) = \mathbf{W} + \Delta\mathbf{W}, \ \Delta\mathbf{W} = \left[(1 - (\mathbf{D}_{:,1})^T\mathbf{W}_{:,1})\mathbf{W}_{:,1}, \ \cdots, \ (1 - (\mathbf{D}_{:,M})^T\mathbf{W}_{:,M})\mathbf{W}_{:,M}\right]$$

With these formulas, we are able to write down the PGD, which is listed in Algorithm 1.

---

**Algorithm 1:** Projected gradient descent for solving (16)

---

**Input:** Dictionary $\mathbf{D} \in \mathbb{R}^{N \times M}$.
**Initialize:** Let $\mathbf{W}^0 = \mathbf{D}$.
**1 for** $j = 0, 1, 2, \ldots$ *until convergence* **do**
**2** $\quad$ Update $\mathbf{W}$ by $\mathbf{W}^{j+1} = \text{Proj}_{\mathcal{W}}\left(\mathbf{W}^j - \eta\mathbf{D}(\mathbf{D})^T\mathbf{W}^j\right)$.
**3 end**
**Output:** $\mathbf{W}^J$, where $J$ is the last iterate.

---

In each step, calculating the gradient has the complexity of $O(N^2M)$ because $\mathbf{D}\mathbf{D}^T$ can be pre-computed. Calculating the projection takes $O(NM)$ time consumptions. Due to the objective function to minimize in (16) is restricted strongly convex, Algorithm 1 is linear convergent (Zhang & Cheng, 2015). To get an $\epsilon$-accurate solution, PGD takes $O(\log(1/\epsilon))$ steps. Thus, the complexity of Algorithm 1 is $O(\log(1/\epsilon)N^2M)$. We should note that the bounds given in Table 1 are the number of parameters to train, not the training complexity. The training complexity can be estimated by "Number of iterations $\times$ complexity of back-propagation", i.e., $O(IBKNM)$, where $I$ is the number of iterations for training, $B$ is the batch size , and $K$ is the number of layers. Actually, Algorithm 1 (Stage 1) only takes a few seconds on an example of $\mathbf{D} : 250 \times 500$, while the training process (Stage 2) of, for example, ALISTA, takes around 0.1 hours.

### E.2 AN EFFICIENT ALGORITHM TO SOLVE (24)

In this section, we introduce an algorithm to solve (24) (we copy (24) below to facilitate reading):

$$\min_{\substack{\mathbf{w} \in \mathbb{R}^{D^2M} \\ \mathbf{w}_m \cdot \mathbf{d}_m = 1, \ 1 \le m \le M}} \left\|\left(\mathbf{W}_{\text{cir}}^N(\mathbf{w})\right)^T\mathbf{D}_{\text{cir}}^N(\mathbf{d})\right\|_F^2.$$

Similarly, by (58), the above problem is equivalent with

$$\min_{\substack{\mathbf{w} \in \mathbb{R}^{D^2 M} \\ \mathbf{d}_m \cdot \mathbf{w}_m = 1, \ 1 \leq m \leq M}} \left\| \left(\mathbf{D}_{\text{cir}}^N(\mathbf{d})\right)^T \mathbf{W}_{\text{cir}}^N(\mathbf{w}) \right\|_F^2. \tag{59}$$

Since the circular convolution is very efficient to calculate in the frequency domain, we consider solving (59) utilizing the fast Fourier transform (FFT).

Firstly, we introduce the operators $\mathbf{D}_{\text{cir}}^N(\mathbf{d}), \mathbf{W}_{\text{cir}}^N(\mathbf{w})$ in the frequency domain. To simplify the notation, we denote the operators as $\mathbf{D}_{\text{cir}}^N$ and $\mathbf{W}_{\text{cir}}^N$ respectively. Let $\mathcal{F}$ be the FFT operator. Thus, $\mathbf{b} = \mathbf{D}_{\text{cir}}^N \mathbf{x}$ is equivalent with

$$\mathcal{F}\mathbf{b} = \mathcal{F}\mathbf{D}_{\text{cir}}^N \mathcal{F}^H \mathcal{F}\mathbf{x}.$$

Let $\hat{\mathbf{b}} = \mathcal{F}\mathbf{b}, \hat{\mathbf{x}} = \mathcal{F}\mathbf{x}$ be the frequency domain signals, let $\hat{\mathbf{D}}_{\text{cir}}^N = \mathcal{F}\mathbf{D}_{\text{cir}}^N \mathcal{F}^H$ be the frequency domain operator. The above equation is:

$$\hat{\mathbf{b}} = \hat{\mathbf{D}}_{\text{cir}}^N \hat{\mathbf{x}}.$$

The frequency domain operator $\hat{\mathbf{D}}_{\text{cir}}^N$ is much cheaper to calculate than the operator $\mathbf{D}_{\text{cir}}^N$ in the spacial domain because it is block diagonal (Wohlberg, 2016). Specifically, we zero pad $\mathbf{d}$ to $N \times N$ and do FFT: $\hat{\mathbf{d}}_m = \text{FFT}\big(\text{zeropad}(\mathbf{d}_m, N - D)\big)$, then the above operator can be explicitly written as:

$$\hat{\mathbf{b}} = \sum_{m=1}^M \overline{\hat{\mathbf{d}}_m} \odot \hat{\mathbf{x}}_m,$$

where $\bar{\cdot}$ means complex conjugate. This is due to $\mathbf{D}_{\text{cir}}^N$ is actually cross-correlation, not convolution (see (18)). Cross-correlation is equal to the transpose of convolution. Thus, there should be complex conjugate in the frequency domain.

Further, since

$$\left\| \left(\hat{\mathbf{D}}_{\text{cir}}^N\right)^H \hat{\mathbf{W}}_{\text{cir}}^N \right\|_F^2 = \left\| \left(\mathcal{F}\mathbf{D}_{\text{cir}}^N \mathcal{F}^H\right)^H \mathcal{F}\mathbf{W}_{\text{cir}}^N \mathcal{F}^H \right\|_F^2 = \left\| \mathcal{F}\left(\mathbf{D}_{\text{cir}}^N\right)^T \mathbf{W}_{\text{cir}}^N \mathcal{F}^H \right\|_F^2$$
$$= \left\| \left(\mathbf{D}_{\text{cir}}^N\right)^T \mathbf{W}_{\text{cir}}^N \right\|_F^2,$$

problem (59) is equivalent with

$$\min_{\substack{\mathbf{w} \in \mathbb{R}^{D^2 M} \\ \mathbf{d}_m \cdot \mathbf{w}_m = 1, \ 1 \leq m \leq M}} \left\| \left(\hat{\mathbf{D}}_{\text{cir}}^N\right)^H \hat{\mathbf{W}}_{\text{cir}}^N \right\|_F^2,$$

which can be efficiently solved by the frequency domain ISTA in (Liu et al., 2017). The details are outlined in Algorithm 2.

## F    VISUALIZATION OF THE ANALYTIC CONVOLUTIONAL WEIGHTS

Fig. 6 visualizes the dictionary $\mathbf{d}$ ($7 \times 7 \times 64$) and the weights $\tilde{\mathbf{w}} \in \mathcal{W}_{\text{cir}}^{13}$, used in the convolutional A-LISTA simulation of Section 5.2. It is obtained by Algorithm 2 in Appendix E.2.

## G    ALGORITHM DETAILS OF TRAINING ROBUST ALISTA

### G.1    MODEL ARCHITECTURE

Inspired by the a similar unrolling and truncating fashion in LISTA, we can approximately solve the coherence minimization problem (16) using a similar finite-layer neural network that is unfolded from iterative algorithms. Because the linear constraints in (16) are hard to enforce in deep neural networks, we first relax it to the following form:

$$\arg\min_{\mathbf{W} \in \mathbb{R}^{N \times M}} \left\| \mathbf{Q} \odot \left(\mathbf{D}^T \mathbf{W} - \boldsymbol{I}_M\right) \right\|_F^2, \tag{60}$$

---

**Algorithm 2:** Frequency-domain ISTA for solving (24)

---

**Input:** Dictionary $\mathbf{d} = [\mathbf{d}_1, \cdots, \mathbf{d}_M]^T$, $\mathbf{d}_m \in \mathbb{R}^{D^2}$, $m = 1, 2, \cdots, M$.
**Initialize:** Let $\mathbf{w}^0 = \mathbf{d}$.

1 **for** $j = 0, 1, 2, \ldots$ *until convergence* **do**
2     Zeropad and FFT:

$$\hat{\mathbf{w}}_m^j = \text{FFT}\Big(\text{zeropad}\big(\mathbf{w}_m^j, N - D\big)\Big), \quad m = 1, \cdots, M.$$

3     Compute frequency domain gradient:

$$(\nabla f)_m = \Big( \sum_{m=1}^{M} \hat{\mathbf{d}}_m \odot \bar{\hat{\mathbf{d}}}_m \Big) \odot \hat{\mathbf{w}}_m^j, \quad m = 1, \cdots, M,$$

    where $\bar{\cdot}$ represents the conjugate of a complex number.
4     Compute the next iterate:

$$\mathbf{w}_m^{j+1} = \text{Proj}_{\mathcal{W}_{\text{normal}}}\Big(\text{IFFT}\big(\hat{\mathbf{w}}_m^j - \eta(\nabla f)_m\big)\Big), \quad m = 1, \cdots, M,$$

    where the set $\mathcal{W}_{\text{normal}}$ is defined in (53).
5 **end**
**Output:** $\mathbf{w}^J$, where $J$ is the last iterate.

---

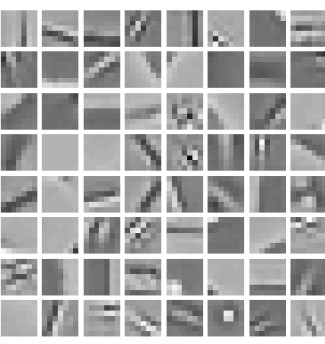

(a) The dictionary $\mathbf{d}$.

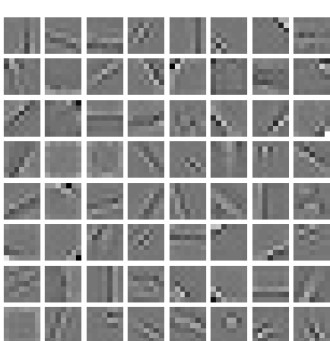

(b) The $\mathbf{w}$ obtained by solving (24).

Figure 6: A visualization of convolutional kernels $\mathbf{d}$ and $\tilde{\mathbf{w}}$, which is obtained by Algorithm 2 and used in the convolutional A-LISTA. $\tilde{\mathbf{w}}$ keeps the high-frequency texture in $\mathbf{d}$. The support of $\mathbf{w}$ is small, most of the pixels in $\mathbf{w}$ are zeros. Then the coherence between shifted $\mathbf{d}$ and $\mathbf{w}$ is nearly 0.

where $\odot$ is the Hadamard product and $\mathbf{Q}$ is a weight matrix that put more penalty on errors on diagonals, because entries on the diagonal will be far smaller than off-diagonal. The above relaxed coherence minimization can be solved using the gradient descent algorithm:

$$\mathbf{W}^{(k+1)} = \mathbf{W}^{(k)} - \gamma^{(k)}\mathbf{D}(\mathbf{Q}^2 \odot (\mathbf{D}^T\mathbf{W}^{(k)} - \boldsymbol{I}_M)). \tag{61}$$

By unfolding (61) and truncate to $K$ steps, and considering the $\gamma^{(k)}\mathbf{D}^T$ outside the residual as learnable parameters $\mathbf{B}$, we will have a deep neural network $\mathbf{W} = E(\mathbf{D})$ as a coherence minimizer. We call it a Stage 1 *encoder* as it encodes a dictionary $\mathbf{D}$ into a weight matrix, that can be used in the Stage 2 of ALISTA, refered as a *decoder*. One layer of this model is shown in Fig. 7(a).

The illustration of the whole feed-forward robust model is shown in Fig. 7(b). The two parts, the encoder and the decoder, can be jointly trained to gain the most from data-driven learning. We further adopt pre-training and curriculum learning to stabilize training, as to be discussed below.

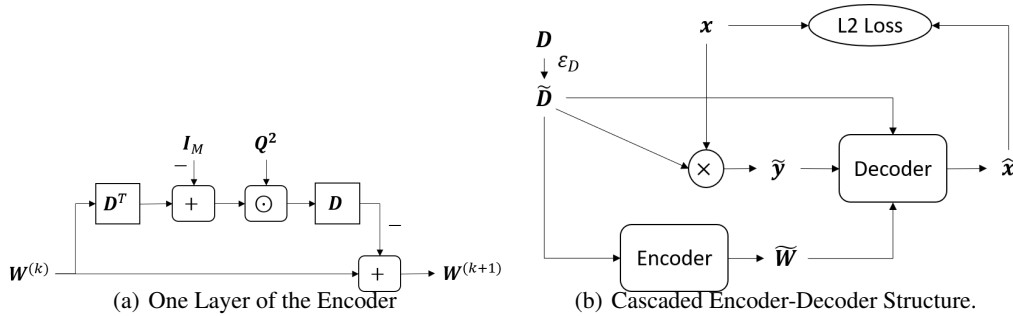

(a) One Layer of the Encoder

(b) Cascaded Encoder-Decoder Structure.

Figure 7: Feed-Forward Analytic LISTA.

## G.2 MODEL TRAINING

To stabilize the training process, we train the model in two stages: the pre-training stage and curriculum (joint) training stage.

**Pre-Training Stage.** We first pre-train the encoder and the decoder individually. The pre-training of the decoder, e.g., ALISTA, follows the standard training procedure in Section 5.1, without bothering the perturbations of $D$. On the other hand, the encoder will always see perturbed dictionaries $\tilde{D} = D + \varepsilon_D$, where $\varepsilon_D$'s entries are sampled from i.i.d. normal distribution with zero mean and $\sigma_{pre}^2$ variance, and update its weight to minimize loss function defined by (60). The $\sigma_{pre}$ is a hyperparameter that we manually select for the pre-training stage, with a default value of 0.01. We use an exponentially decaying learning rate for encoder pre-training with an initial value $\alpha_{pre} = 10^{-4}$.

**Curriculum (Joint) Training Stage.** After the pre-training stage, we concatenate these two parts and do joint training. However, a direct end-to-end tuning was observed to cause much instability, due to the randomness in weights. Inspired by the curriculum learning technique, we first perturb the dictionaries with smaller standard deviations and gradually increase the perturbation level during training. Specifically, starting from a small standard deviation $\sigma^t = \sigma^0$, the curriculum joint training procedure repeats the routine below:

- First uniformly sample a batch of standard deviations $\{\sigma_i\}_{i=1}^{B_D}$ from $[0, \sigma^t]$, where $B_D$ is the batch size for perturbations of the original dictionary $D$. Use the sampled standard deviations to sample $B_D$ perturbations and apply to $D$ and then normalized them to get $\{\tilde{D}\}_{i=1}^{B_D}$.

- Sample a batch of sparse codes $\{x_j\}_{j=1}^{B_x}$ from a pre-defined Gaussian-Bernoulli distribution; the supports of the sparse codes are decided i.i.d. by a Bernoulli distribution to have around $10\%$ non-zero entries; and the magnitudes are sampled from i.i.d. standard Gaussian. $B_x$ is the batch size for sparse codes in training.

- Measure $y_{i,j} = \tilde{D}_i x_j$. Then $(y_{i,j}, x_j, \tilde{D}_i)$ forms a tripelet of training sample. Note that only Robust ALISTA needs $\tilde{D}_i$ as part of the training samples.

- Feed in the data and update the encoder and decoder with learning rates $\alpha_e$ and $\alpha_d$, using the Adam Optimizer, respectively.

- Increase $\sigma^t$ to the next larger value, after $C$ training batches.

- Repeat the above steps, until the value of $\sigma^t$ exceeds the pre-defined $\sigma_{max}$, that represents the maximal standard deviation to sample the dictionary perturbation.

In the experiment, we have $B_D = 4$, $B_x = 16$, $C = 50000$, $\alpha_e = 10^{-6}$, $\alpha_d = 10^{-4}$ and $\sigma_{max} \in \{0.02, 0.03\}$ and $\sigma^t$ is obtained by linearly interpolating between $[0, \sigma_{max}]$ for $L - 1$ times. We choose $L = 5$; hence $\sigma^t$ takes $\frac{i}{5}\sigma_{max}, i = 1, 2, \ldots, 5$ in order.

## H    RESULTS OF NATURAL IMAGE DENOISING USING CONV ALISTA

The natural image denoising experiment is conducted on the same BSD 500 dataset using the 400-image training set, 50-image validation set and 50-image test set. We convert them all to grayscale, and then add $\sigma = 20$ Gaussian i.i.d. noise. We train both Conv LISTA (i.e., model (20)) and Conv ALISTA (i.e., model (26)). Both networks have 5 layers, with the same dictionary $\mathbf{D}$ obtained from the training set by solving (24). We reconstruct the denoised images using by convolving the learned feature maps with the original dictionary $\mathbf{D}$. The mean-square-error (MSE) between denoised and clean images are adopted as the network training loss, as inspired by (Zhou et al., 2018).

Six popular benchmark images (adding $\sigma = 20$ noise) are tested and reported in Table 4. The *A-PSNR* denotes the average PSNR over all images and the *A-Times* represents the average inference time (in seconds) for denoising one image. We compare Conv LISTA and Conv ALISTA, as well as the classical KSVD denoising algorithm (Elad & Aharon, 2006) and the recent CSC denoising algorithm with gradient regularization (CSC-GR) (Wohlberg, 2018). The results show that Conv LISTA and Conv ALISTA (without heavy tuning done for their optimal performance) can perform comparably with KSVD and outperforms CSC-GR, but with tremendously faster inference speeds than KSVD/CSC-GR. More importantly, Conv LISTA and Conv ALISTA only have marginal performance differences, validating again the analytic weights in convolutional cases.

Table 4: Peak Signal to Noise Ratio (PSNR) Comparision between Conv LISTA and Conv ALISTA.

| Model | Image PSNR (dB) | | | | | | A-PSNR | A-Time |
|---|---|---|---|---|---|---|---|---|
| | Lenna | House | Pepper | Couple | Boats | Barbara | | |
| KSVD | 31.03 | 33.24 | 30.97 | 31.71 | 31.00 | 30.47 | 31.40 | 24.70 |
| CSC-GR | 28.41 | 29.11 | 27.39 | 29.31 | 28.35 | 27.19 | 28.29 | 7.56 |
| Conv LISTA | 31.26 | 32.77 | 31.00 | 31.89 | 30.78 | 29.53 | 31.21 | 0.012 |
| Conv ALISTA | 31.01 | 32.46 | 30.81 | 31.85 | 30.58 | 29.72 | 31.07 | 0.014 |

## I    RESULTS OF ABLATION STUDIES IN ROBUSTNESS EXPERIMENTS

As one anonymous reviewer kindly pointed out, Robust ALISTA has larger parameter space over TiLISTA and ALISTA trained. Therefore, they suggested that we increased the number of layers in TiLISTA, ALISTA and the baseline model LISTA-CPSS in (Chen et al., 2018) to see if their performance in this above evaluation setting can be improved in that way. Note that LISTA-CPSS has tens of layers, hence actually containing more parameters than robust ALISTA. In addition, the reviewers also suggested a set of ablation studies to investigate whether more layers in the encoder can endorse the model better adaptivity to higher level perturbations. We conduct the suggested experiments and present the results in this section.

### I.1    NUMBER OF LAYERS IN TiLISTA, ALISTA AND LISTA-CPSS

As the reviewers pointed out, the comparison we present in Section. 5.3 might be unfair because robust ALISTA contains much more parameters (because it contains a 4-layer encoder) comparing to ALISTA, which only learns two series of scalars, and TiLISTA which has just one more matrix weight than ALISTA. Therefore, we add the following experiments to consolidate our claim on the effectiveness of the robust ALISTA model:

- we increase the number of layers of TiLISTA and ALISTA which are then trained in the same data augmentation setting as we do in Section. 5.3, to see if they could yield comparitive robustness against dictionary perturbations;

- we also compare the robust ALISTA with the baseline LISTA-CPSS model in Chen et al. (2018), which contains tens of layers of independent weight matrices, thus having even more parameters than robust ALISTA. This comparison could consolidate our claim that the outstanding adaptiveness to dictionary perturbations of robust ALISTA is brought by its encoder-decoder structure rather than its learning capacity alone.

The results are shown in Table. 5, where the performances are measured with NMSE in dB, which is defined in Section 5.1. The "Augmented" prefix means the models are trained in the data augmentation setting. $\sigma$ is the standard deviation of the Gaussian distribution that is used to generate the dictioanry perturbations. $T$ stands for the number of layers (in the case of robust ALISTA it means the nubmer of layers of the ALISTA decoder, with a 4-layer encoder). We follow the training strategy and settings explained in Appendix G, with $\sigma_{max} = 0.02$ during training.

On one hand, the comparison of performances of ALISTA, TiLISTA and LISTA-CPSS shows results that are consistent to the intuition that larger parameter space yields larger learning capacity, and therefore, better adaptiveness (LISTA-CPSS > TiLISTA > ALISTA). On the other hand, we can also find that ALISTA with more layers has worse performance. We think this observation is also reasonable for two reasons: 1) adding more layers in ALISTA does not enlarge the parameter volume significantly because it has only two scalar parameters in each layer; noting that ALISTA uses a fixed, analytically solved weight matrix, if this weight matrix is not compatible with the perturbed dictionary, more layers can even hurt the performance instead of improving. Lastly, it's clearly shown that robust ALISTA outperforms LISTA-CPSS, even if it contains less parameters. This proves that the encoding process that adaptively transforms the perturbed dictiories is necessary to achieve good robustness against perturbations in dictionaries.

| $\sigma$ of perturbations during testing | | 0.0001 | 0.001 | 0.01 | 0.015 | 0.02 | 0.025 |
|---|---|---|---|---|---|---|---|
| Augmented ALISTA | T=16 | -26.58 | -25.87 | -15.49 | -11.71 | -8.84 | -6.74 |
| | T=20 | -24.43 | -24.46 | -15.39 | -11.77 | -8.94 | -6.82 |
| | T=24 | -24.12 | -24.00 | -15.45 | -11.68 | -8.81 | -6.70 |
| Augmented TiLISTA | T=16 | -27.76 | -27.18 | -16.83 | -12.95 | -9.81 | -7.55 |
| | T=20 | -28.13 | -28.54 | -17.15 | -12.98 | -9.83 | -7.58 |
| | T=24 | -26.08 | -27.27 | -17.34 | -13.14 | -9.91 | -7.61 |
| Augmented LISTA-CPSS | T=16 | -27.93 | -27.18 | -16.96 | -12.99 | -9.93 | -7.70 |
| | T=20 | -28.17 | -27.33 | -16.95 | -13.00 | -9.94 | -7.71 |
| | T=24 | -30.30 | -29.24 | -16.86 | -12.97 | -9.94 | -7.70 |
| Robust ALISTA | T=16 | -62.47 | -62.41 | -62.02 | -61.50 | -60.67 | -45.00 |

Table 5: The results (recovery NMSE in dB) of ablation study on the influence of model capacity towards the model robustness against dictionary perturbations.

## I.2  Ablation Study on the Depths of Encoders in Robust ALISTA

Another constructive suggestion from the reviewers is to design an ablation study to investigate the influence of the depth of encoders in robust ALISTA on its adaptivity to dictionary perturbations. A natural intuition is that, adding more layers to the encoder can increase its ability to sustain larger perturbation levels. But is this true?

Therefore, we train another two robust ALISTA models, with a 5-layer and a 6-layer encoders respectively and 16-layer ALISTA decoders for both, and compare them with the originally reported robust ALISTA model with a 4-layer encoder and a 16-layer ALISTA decoders. All three models use one pretrained decoder, and pretrain their encoders using the same method (see Appendix G). We only use $\sigma_{max} = 0.02$ during training. One thing to notice is that we observe unstable training process if we use default initial learning rates $\alpha_{pre} = 10^{-4}$ in the pre-training stage and $\alpha_e = 10^{-6}$ in the joint training stage when encoders have 5 or 6 layers. Therefore, we use $\alpha'_{pre} = 10^{-5}$ for the 6-layer encoder in the pre-training stage, and in the joint training stage use a decreased and uniform initial learning rate $\alpha'_e = 10^{-9}$ for the three encoders while keeping the default initial learning rate $\alpha_d = 10^{-4}$ for decoders. The other settings remain the same.

Results are shown in Table. 6. The performances are measured with NMSE in dB, defined in Section 5.1. From the table we can see that encoders do show better robustness when they have more layers, i.e. larger learning capacity.

| # Encoder Layers | $\sigma$ of perturbations during testing | | | | | |
|:---:|:---:|:---:|:---:|:---:|:---:|:---:|
| | 0.0001 | 0.001 | 0.01 | 0.015 | 0.02 | 0.025 |
| 4 | -68.57 | -68.56 | -67.94 | -66.86 | -64.84 | -56.63 |
| 5 | -69.34 | -69.34 | -69.02 | -68.49 | -67.20 | -65.55 |
| 6 | -70.38 | -70.33 | -69.92 | -69.22 | -67.72 | -65.60 |

Table 6: The results of ablation study on the influence of the depths of encoder towards the model robustness against dictionary perturbations.

