# OpenReview forum: "ALISTA: Analytic Weights Are As Good As Learned Weights in LISTA"
_ICLR.cc/2019/Conference_

### Official Review · AnonReviewer3 · 2018-10-20
**ALISTA: Analytic Weights Are As Good As Learned Weights in LISTA**

**Rating:** 9
**Confidence:** 5

**Review:**

The papers studies neural network-based sparse signal recovery, and derives many new theoretical insights into the classical LISTA model. The authors proposed Analytic LISTA (ALISTA), where the weight matrix in LISTA is pre-computed with a data-free coherence minimization, followed by a separate data-driven learning step for merely (a very small number of) step-size and threshold parameters. Their theory is extensible to convolutional cases. The two-stage decomposed pipeline was shown to keep the optimal linear convergence proved in (Chen et al., 2018). Experiments observe that ALISTA has almost no performance loss compared to the much heavier parameterized LISTA, in contrast to the common wisdom that (brutal-force) “end-to-end” always outperforms stage-wise training. Their contributions thus manifest in both novel theory results, and the practical impacts of simplifying/accelerating LISTA training.  Besides, they also proposed an interesting new strategy called Robust ALISTA to overcome the small perturbations on the encoding basis, which also benefits from this decomposed problems structure.

The proofs and conclusions are mathematically correct to my best knowledge. I personally worked on similar sparse unfolding problems before so this work looks particularly novel and interesting to me. My intuition then was that, it should not be really necessary to use heavily parameterized networks to approximate a simple linear sparse coding form (LISTA idea). Similar accelerations could have been achieved with line search for something similar to steepest descent (also computational expensive, but need learn step-sizes only, and agnostic to input distribution). Correspondingly, there should exist a more elegant network solution with very light learnable weights. This work perfectly coincides with the intuition, providing very solid guidance on how a LISTA model could be built right. Given in recent three years, many application works rely on unfold-truncating techniques (compressive sensing, reconstruction, super resolution, image restoration, clustering…), I envision this paper to generate important impacts for practitioners pursuing those ideas.

Additionally, I like Theorem 3 in Section 3.1, on the provable efficient approximation of general convolution using circular convolution. It could be useful for many other problems such as filter response matching.

I therefore hold a very positive attitude towards this paper and support for its acceptance. Some questions I would like the authors to clarify & improve in revision:

1.	Eqn (7) assumes noise-free case. The author stated “The zero-noise assumption is for simplicity of the proofs.” Could the authors elaborate which part of current theory/proof will fail in noisy case? If so, can it be overcome (even by less “simpler” way)? How about convolutional case, the same? Could the authors at least provide some empirical results for ALISTA’s performance under noise?

2.	Section 5.3. It is unclear to me why Robust ALISTA has to work better than the data augmented ALISTA. Is it potentially because that in the data augmentation baseline, the training data volume is much amplified, and one ALISTA model might become underfitting? It would be interesting to create a larger-capacity ALISTA model (e.g., by increasing unfolded layer numbers), train it on the augmented data, and see if it can compare more favorably against Robust ALISTA?

3.	The writeup is overall very good, mature, and easy to follow. But still, typos occur from time to time, showing a bit rush. For example, Section 5.1, “the x-axes denotes is the indices of layers” should remove “is”. Please make sure more proofreading will be done.

---

> ### Author Response · Authors · 2018-11-13
> **Response to Reviewer 3**
>
> Thank you for your careful reading and comments!
>
> - Q1: In our proofs, we take b as b=Ax*. If we add noise to the measurements, almost all the inequalities in the proof need to be modified. We will end up getting “convergence” to a neighbor of x* with a size depending on the noise level. Such modifications also apply to the analysis for convolutional dictionaries. Numerically, figures 1(b), 1(c) and 1(d) depict the results of ALISTA under SNRs = 40dB, 30dB and 20dB, respectively.
>
> - Q2: We basically agree with your comment on why data augmented TiLISTA and ALISTA are not performing as well as robust ALISTA. We are conducting the experiments that you have suggested and will update the results in comments once they become available, and also add them to the paper’s next update.
>
> - Q3: Thanks for kindly pointing out our writing issues. We will carefully fix typos and use more proofreading.

---

> ### Author Response · Authors · 2018-11-15
> **Continued Response to Reviewer 3**
>
> As you kindly suggested, we added two experiments to train the data-augmented version of ALISTA with 20 and 24 layers, to compare with the robust ALISTA model (the concatenation of a feed-forward encoder network that learns to solve the coherence minimization and a ALISTA network with step size and thresholds parameters).
>
> For training ALISTA with data-augmentation, in each step, we first generate a batch of perturbed dictionaries \tilde{D}s around an original dictionary D. Then these perturbed dictionaries are used to generate observations, by multiplying sparse vector samples from the same distribution. The data-augmented version of ALISTA is then trained with those dictionary-perturbed samples. It still follows the standard ALISTA to use a fixed weight matrix W that is analytically pre-solved from the original dictionary D.
>
> The robust ALISTA model instead uses the encoder network to adaptively produce weight matrices to be used in ALISTA. Apart from the encoder network, the robust ALISTA needs to learn a set of step size and thresholds parameters just like the baseline ALISTA. We fix using a 16-layer ALISTA network and a 4-layer encoder in the robust ALISTA model.
>
> In this experiment, we  compare both models’ robustness to dictionary perturbations, by plotting recovery normalized MSEs (in dB) in testing, w.r.t. the standard deviation of perturbation noise, and also w.r.t. the layers used for data-augmented ALISTA. We set the maximal standard deviation of generated perturbations to 0.02 and followed the same settings described in Appendix E in the paper:
>
> Sigma (standard deviation)   |   0.0001    |    0.001    |     0.01      |    0.015    |    0.02     |   0.025
> Augmented ALISTA T=16       |   -26.58     |    -25.87   |   -15.49    |   -11.71   |   -8.84    |   -6.74
> Augmented ALISTA T=20       |   -24.43     |   -24.46    |   -15.39    |   -11.77   |   -8.94    |   -6.82
> Augmented ALISTA T=24       |   -24.12     |   -24.00    |   -15.45    |   -11.68   |   -8.81    |   -6.70
> Robust ALISTA T=16                |   -62.47     |   -62.41    |   -62.02    |   -61.50   |   -60.67  |   -45.00
>
>
> - Observation: as we may see in the above results, more layers didn’t bring obvious empirical benefits to the recoverability of ALISTA. We could even observe that ALISTA of 24 layers had slightly worse NMSE that ALISTA of 16 and 20 layers.
>
> - Analysis: we agree with your insight that the limited parameter volume of augmented ALISTA  might limited its capacity and robustness to recover from dictionary-perturbed measurements, compared to robust ALISTA which has another encoder network that adaptively and efficiently encodes the perturbed dictionary \tilde{D} into new (dynamic) weight matrix \tilde{W}. ALISTA only has two scalars to be learned in each layer (one scalar as step size and the other as threshold), therefore adding more layers do not enlarge the parameter volume significantly.
>
> - Remark: from the comparison, we could conclude that it takes more than adjusting step sizes and thresholds to gain robustness to dictionary perturbations in LISTA/ALISTA. Therefore, robust ALISTA makes the meaningful progress in creating an efficient encoder network, that can dynamically address the dictionary variations \tilde{D} by always adjusting \tilde{W}. Without incurring much higher complexity, robust ALISTA witness remarkable improvements over ALISTA, making it a worthy effort in advancing LISTA-type network research into the practical domain.

---

### Official Review · AnonReviewer2 · 2018-10-29
**ALISTA - Review**

**Rating:** 7
**Confidence:** 4

**Review:**

The paper describes ALISTA, a version of LISTA that uses the dictionary only for one of its roles (synthesis) in ISTA and learns a matrix to play the other role (analysis), as seen in equations (3) and (6). The number of matrices to learn is reduced by tying the different layers of LISTA together.

The motivation for this paper is a little confusing. ISTA, FISTA, etc. are algorithms for sparse recovery that do not require training. LISTA modified ISTA to allow for training of the "dictionary matrix" used in each iteration of ISTA, assuming that it is unknown, and offering a deep-learning-based alternative to dictionary learning. ALISTA shows that the dictionary does not need to change, and fewer parameters are used than in LISTA, but it still requires learning matrices of the same dimensionality as LISTA (i.e., the reduction is in the constant, not the order). If the argument that fewer parameters are needed is impactful, then the paper should discuss the computational complexity (and computing times) for training ALISTA vs. the competing approaches.

There are approaches to sparse modeling that assume separate analysis and synthesis dictionaries (e.g., Rubinstein and Elad, "Dictionary Learning for Analysis-Synthesis Thresholding"). A discussion of these would be relevant in this paper.

* The intuition and feasibility of identifying "good" matrices (Defs. 1 and 2) should be detailed. For example, how do we know that an arbitrary starting W belongs in the set (12) so that (14) applies?
* Can you comment on the difference between the maximum entry "norm" used in Def. 1 and the Frobenius norm used in (17)?
* Definition 3: No dependence on theta(k) appears in (13), thus it is not clear how "as long as theta(k) is large enough" is obtained.
* How is gamma learned (Section 2.3)?
* The notation in Section 3 is a bit confusing - lowercase letters b, d, x refer to matrices instead of vectors. In (20), Dconv,m(.) is undefined; later Wconv is undefined.
* For the convolutional formulation of Section 3, it is not clear why some transposes from (6) disappear in (21).
* In Section 3.1, "an efficient approximated way" is an incomplete sentence - perhaps you mean "an efficient approximation"?. Before (25), Dconv should be Dcir? The dependence on d should be more explicitly stated.
* Page 8 typo "Figure 1 (a) (a)".
* Figure 2(a): the legend is better used as the label for the y axis.
* I do not think Figure 2(b) verifies Theorem 1; rather, it verifies that your learning scheme gives parameter values that allow for Theorem 1 to apply (which is true by design).
* Figure 3: isn't it easier to use metrics from support detection (false alarm/missed detection proportions given by the ALISTA output)?

---

> ### Author Response · Authors · 2018-11-13
> **Response to Reviewer 2 (Continued)**
>
> Answers to individual comments:
>
> - Q1 (Intuition and feasibility of identifying "good" matrices; Definition 1):
> Definition 1 describes a property of good matrices: small coherence with respect to D. This is inspired by Donoho & Elad, 2003; Elad, 2007; Lu et al, 2018. Our Theorem 1 validates this point: a small mutual coherence leads to a large c and faster convergence. Feasibility is proved in (Chen et al., 2018). We have added these clarifications in our update.
>
> - Q1 (Clarification of Definition 2):
> Because W and D are both “fat” matrices, the product W’D, and such products of their submatrices consisting of two or more their corresponding columns, generally cannot be very close to the identity matrix. For a given D, Definition 2 let sigma_min represent the minimal “distance” and define the set of corresponding W matrices. A larger sigma_min implies slower convergence in Theorem 2.  We have added numerical validations of (11) to the appendix in the update. (The original definition (12) is (11) in the updated version.)
>
> - Q2 (Difference between the maximum entry "norm" and the Frobenius norm):
> We use a Frobenius norm in (16) instead of a sup-norm in Def. 1 (8) for computational efficiency. Directly minimizing the sup norm leads to a large-scale linear program. The sizes of the matrices W and D that we used in our numerical experiments are 250 by 500. We implemented an LP solver for the sup-norm minimization (8) based on Gurobi, which requires more than 8GB of memory and may be intractable on a typical PC. However, solving (16) in MATLAB needs only around 10MB of memory and a few seconds. Besides the Frobenius norm, we also tried to minimize the L_{1,1} norm but found no advantages. (The original formula (17) is (16) in the update.)
>
> - Q3 (Definition 3):
> By (6), x^k depends on thresholding parameters theta^0, theta^1, ..., theta^{k-1}. When these theta parameters are large enough, x^k can be sufficiently sparse. Theorem 1 implies we can ensure “support(x^k) belongs to S” for all k by properly choosing the theta^k sequence.
>
> - Q4 (How is gamma learned):
> The step sizes gamma^k and thresholds theta^k (for all k) are updated to minimize the empirical recovery loss in (5), using the standard training method based on backpropagation and the Adam method. For ALISTA, the big Theta in (5), which is the set of parameters subject to learning, consists of only gammas and thetas. The matrix W is pre-computed by analytic optimization and, therefore, is fixed during training.
>
> - Q5 (The notation in Section 3):
> The lowercase letters are always vectors. The matrices D_{conv,m} are defined so that (18), which is precise but complicated, is equivalent to (19), which is simple and compact. The full definition of D_{conv,m} is given in Appendix C.2. The matrices W_{conv,m} are defined for a similar purpose before (21). We have added these clarifications in the updated version. (The original formula (20) is (19) in the current version.)
>
> - Q6 (Transpose in convolution):
> Transposing a circulant matrix is equivalent to applying the convolution with rotated filters (Equation (6) and Footnote 2 in Chalasani et al., 2013). We have made clarifications in the update.
>
> - Q7 & Q8 & Q9 (Typos and figure suggestions):
> Thanks for finding the typos and making suggestions for figures. We have fixed the typos and will carefully proofread our paper.
>
> - Q10 (“I do not think Figure 2(b) verifies Theorem 1”):
> We agree that we incorrectly used the words "verify" and "validation." Rather, the numerical observations in Figure 2(b) justify our choices of parameters in Theorem 1. We have made this correction.
>
> - Q11 (Figure 3):
> We agree that the number and proportion of false alarms are a more straightforward performance metric. However, they are sensitive to the threshold. We found that, although using a smaller threshold leads to more false alarms, the final recovery quality is better and those false alarms have small magnitudes and are easy to remove by thresholding during post-processing. That's why we chose to show their magnitudes, implying that we get easy-to-remove false alarms. We have added this reasoning to the final version.

---

> > ### Comment · AnonReviewer2 · 2018-11-29
> > **Rating has been upgraded**
> >
> > Just some minor comments on the responses from the authors:
> > [Removed comments that were incorrectly included in this response]
> > * The complexity of the algorithm should also include that of the optimization that finds the matrix W, equation (16) in Stage 1.

---

> > > ### Author Response · Authors · 2018-12-04
> > > **Re: Rating has been upgraded**
> > >
> > > [Opening is okay]
> > >
> > > Points 1 and 2: There is no word "tree" or "graph", no "beta" or "$\beta$", in our paper. We are confused and think they may refer to another paper. Could you kindly clarify?
> > >
> > > 3: This is great suggestion. The matrix W is the solution of a convex quadratic program subject to linear constraints and, thereby, a linear system. Solving this system costs a negligible amount compared to training the remaining parameters. For example, when W is 250-by-500, computing W takes a few seconds but the remaining of ALISTA takes 1.5 hours. As you suggested, we will add this explanation and the complexity of computing W to the camera-ready version.

---

> ### Author Response · Authors · 2018-11-13
> **ALISTA pre-computes the weight matrix and only learns a series of threshold and step size parameters**
>
> Thanks for your careful review and the comments! We have revised our paper and we believe our responses and revisions address your concerns. We would be very grateful if you would look over our paper again, and reconsider your opinion.
>
> Let us first provide a general response, followed by responses to your specific comments.
>
> The goal of work is to significantly speed up sparse recovery. The basis of this line of work is ISTA (iterative soft-thresholding algorithm), a classic iterative method for recovering a sparse vector x from it linear measurements Dx, which are further contaminated by additive noise. Like most iterative methods, ISTA repeats the same operation (matrix-vector multiplications by D and D’ and a soft thresholding) at each iteration. Therefore, it can be written as a simple for-loop. However, depending on the problem condition, it can take hundreds of iterations or tens of thousands of iterations. Gregor & LeCun, 2010, instead of using the original matrices D and D’ and soft-thresholding scalars in ISTA, select a series of new matrices and scalars by training using a set of synthetic sparse signals and their linear measurements. The resulting method, called LISTA (or learned ISTA), has a small fixed number of iterations, roughly 20, and is not only much faster but recovers more accurate sparse vectors than ISTA even if ISTA runs order-of-magnitude more iterations. On the other hand, training LISTA takes a long time, typically ten hours or longer, much like training a neural network with lots of parameters. Also, one must train new matrices and scalars for each encoding matrix D. These shortcomings are addressed by a line of work that follows LISTA.
>
> This paper introduces ALISTA, which significantly simplifies LISTA by using only one free matrix (besides the encoding matrix D) for all iterations, and pre-computing that matrix by analytic optimization, as opposed to data-driven training. Therefore, when it comes to training ALISTA, there remain only a series of scalars for thresholding and step sizes to be learned from synthetic data. Despite this huge simplification, the performance of ALISTA is no worse than LISTA and other work along the line, supported by our theoretical results and numerical verification.
>
> Your question on computational complexity is great. Let us compute how much saving in flops ALISTA has over LISTA or its variants. Assume there are K layers (i.e., iterations) in total, and the encoding matrix has N rows and M columns with N < M, possibly N << M. In its typical implementation, vanilla LISTA learns O(KM^2+K+MN) parameters. That is one matrix and one scalar per layer and another matrix shared between all layers. LISTA in Chen et al., 2018 (also (6) in this paper) learns O(KNM + K) parameters as they learn only one N-by-M matrix and one thresholding parameter per layer. Tied LISTA ((15) in this paper) learns only O(NM + K) parameters by using only one matrix for all the K layers plus a step size and a thresholding parameter per layer. ALISTA ((16) in this paper) learns only O(K) parameters because it determines the only matrix by analytic optimization and fixes it during training. All these methods achieve similar recover quality. We have added this comparison to the revised paper.
>
> The model in the paper that you has mentioned, “Dictionary Learning for Analysis-Synthesis Thresholding”, is related to our paper as a special LISTA model with only one layer. We have cited this and related papers (listed below) in Section 1 of our updated version and discussed their contributions.
>
> Yang et al., 2016. “Analysis-Synthesis Dictionary Learning for Universality-Particularity Representation Based Classification.”

---

### Official Review · AnonReviewer1 · 2018-11-01
**important theoretical contribution to unrolling literature**

**Rating:** 10
**Confidence:** 5

**Review:**

The paper raises many important questions about unrolled iterative optimization algorithms, and answers many questions for the case of iterative soft thresholding algorithm (ISTA, and learned variant LISTA). The authors demonstrate that a major simplification is available for the learned network: instead of learning a matrix for each layer, or even a single (potentially large) matrix, one may obtain the matrix analytically and learn only a series of scalars. These simplifications are not only practically useful but allow for theoretical analysis in the context of optimization theory. On top of this seminal contribution, the results are extended to the convolutional-LISTA setting. Finally, yet another fascinating result is presented, namely that the analytic weights can  be determined from a Gaussian-perturbed version of the dictionary. Experimental validation of all results is presented.

My only constructive criticism of this paper are a few grammatical typos, but specifically the 2nd to  last sentence before Sec 2.1 states the wrong thing "In this way, the LISTA model could be further significantly simplified, without little performance loss"
...
it should be "with little".

---

> ### Author Response · Authors · 2018-11-13
> **Response to Reviewer 1**
>
> Thank you for your careful reading and kindly identifying the typos in our paper! We will fix these typos and meticulously proofread our article.

---

### Meta-Review · Area_Chair1 · 2018-12-17
**Solid contribution to unrolled iterative optimization and soft thresholding**

**Confidence:** 5
**Recommendation:** Accept (Poster)

**Metareview:**

This is a well executed paper that makes clear contributions to the understanding of unrolled iterative optimization and soft thresholding for sparse signal recovery with neural networks.